# 3′UTR shortening alleviates miRNA repression of mRNAs critical for muscle stem cell differentiation

Yi Zhu [1,7], Jianshu Wang[1,7], Deng Tong[1,7], Peixuan Jia[2,7], Suli Chen[1,2], Yangyang Li[3], Jiaying Fu [2], Qiming Li[3], Ping Hu [4,5✉], Yu Zhou [3,6✉] & Hong Cheng [1,2✉]

## Abstract

**Alternative polyadenylation (APA) modulates gene expression by altering 3′ untranslated region (3′UTR) length. Although 3′UTR lengthening typically accompanies cell differentiation, we unexpectedly observed preferential APA-mediated 3′UTR shortening events during muscle stem cell (satellite cell, SC) differentiation, coinciding with increased muscle-specific miRNAs (myomiRs) targeting at alternative 3′UTRs. Mechanistically, this shortening primarily results from reduced cleavage factor I (CFI) expression and allows transcripts to escape repression by differentiation-induced myomiRs. Interestingly, perturbation of mRNA 3′UTR shortening of multiple genes impairs myogenic differentiation. Focusing on *Matr3*—a gene linked to muscle disorders—we demonstrate that its APA-miRNA regulatory balance is critical for efficient SC differentiation in vitro. Genetically mutating *Matr3* proximal polyadenylation site (pA site) impaired mouse muscle regeneration in vivo. Together, our findings reveal that APA-mediated 3′UTR shortening counteracts miRNA repression to orchestrate the gene expression program essential for robust muscle regeneration.**

**Keywords** Alternative Polyadenylation; miRNA; *Matr3*; Myogenic Differentiation; Muscle Regeneration
**Subject Categories** Development; RNA Biology; Stem Cells & Regenerative Medicine

## Introduction

Most eukaryotic genes contain multiple polyadenylation sites (pA sites), enabling alternative polyadenylation (APA) as a key regulatory mechanism (Di Giammartino et al, 2011; Fu et al, 2011; Shepard et al, 2011; Hoque et al, 2013; Shi, 2012; Tian and Manley, 2017). APA modulates gene expression by generating isoforms with distinct 3′UTR lengths. These differences in 3′UTR architecture can critically influence mRNA fate—including localization, stability, and translation—through the inclusion or exclusion of regulatory elements such as miRNA binding sites and RNA-binding protein (RBP) recognition motifs (Tian and Manley, 2017; An et al, 2008; Lau et al, 2010; Fu et al, 2018; Berkovits and Mayr, 2015; Spies et al, 2013). Strikingly, APA is tightly regulated across biological contexts: global 3′UTR shortening is a hallmark of proliferating cells, whereas differentiation typically induces 3′UTR lengthening (Fu et al, 2011; Hoque et al, 2013; Mayr and Bartel, 2009; Gallicchio et al, 2023; Sandberg et al, 2008; Lackford et al, 2014; Yang et al, 2022; Kang et al, 2023a). Up to date, most studies have been focused on APA events conforming to this paradigm, leaving deviations, e.g., 3′UTR shortening events during differentiation, poorly characterized.

miRNA-mediated gene regulation represents another critical layer of post-transcriptional control. They bind 3′UTR sequences to repress target gene expression via mRNA destabilization and/or translational inhibition (Bartel, 2018). miRNA expression undergoes dynamic reprogramming during diverse biological processes, including cellular differentiation (Horak et al, 2016; Coolen and Bally-Cuif, 2009; Baumjohann and Ansel, 2013; Kuppusamy et al, 2013). For example, a group of muscle-specific miRNAs (myomiRs, e.g., miR-1, miR-133, miR-206) are sharply induced during muscle differentiation (Horak et al, 2016; Chen et al, 2006; Kim et al, 2006; Dey et al, 2011).

The inclusion or exclusion of miRNA binding sites in alternative 3′UTR isoforms—regulated by APA—could enable context-dependent control of this repression. For instance, *Pax3* mRNA is selectively silenced by miR-206 in quiescent muscle stem cells (QSCs) of limb muscle but not in diaphragm QSCs, due to tissue-specific APA regulation that retains or excludes miR-206 target sites in its 3′UTR (Boutet et al, 2012; De Morree et al, 2019). Despite this paradigm example, how APA and miRNA pathways interact and function cooperatively or antagonistically in biological contexts remains poorly understood.

Myogenesis is a tightly regulated process required for muscle development and regeneration throughout lifetime, driven by

[1]Key Laboratory of RNA Innovation, Science and Engineering, Shanghai Key Laboratory of Molecular Andrology, Shanghai Institute of Biochemistry and Cell Biology, Center for Excellence in Molecular Cell Science, Chinese Academy of Sciences, University of Chinese Academy of Sciences, Shanghai 200031, China. [2]Key Laboratory of Systems Health Science of Zhejiang Province, School of Life Science, Hangzhou Institute for Advanced Study, University of Chinese Academy of Sciences, Hangzhou 310024, China. [3]College of Life Sciences, TaiKang Center for Life and Medical Sciences, RNA Institute, Hubei Key Laboratory of Cell Homeostasis, Wuhan University, Wuhan 430072, China. [4]Key Laboratory of Biological Targeting Diagnosis, Therapy and Rehabilitation of Guangdong Higher Education Institutes, the Fifth Affiliated Hospital of Guangzhou Medical University, Guangzhou 510320, China. [5]Guangzhou Laboratory, 9 North Xingdaohuan Road, Guangzhou 510005, China. [6]Institute of Advanced Studies, Wuhan University, Wuhan 430072, China. [7]These authors contributed equally: Yi Zhu, Jianshu Wang, Deng Tong, Peixuan Jia. ✉E-mail: hup@sibcb.ac.cn; yu.zhou@whu.edu.cn; hcheng@sibcb.ac.cn

satellite cells (SCs) in postnatal skeletal muscle (Sousa-Victor et al, 2022; Fu et al, 2022). In this study, we unexpectedly uncovered preferential APA-mediated 3′UTR shortening coinciding with myomiR induction during SC differentiation. We provide evidence that this active shortening enables many differentiation-critical transcripts to bypass excessive miRNA repression, thereby facilitating efficient differentiation. Using *Matr3*, a muscle disorder-associated gene (Malik and Barmada, 2021), as a paradigm, we demonstrate that APA-mediated 3′UTR shortening counteracts miRNA-mediated repression for robust myogenic differentiation. Strikingly, genetic disruption of *Matr3* 3′UTR shortening in mice impaired muscle regeneration. Collectively, our findings establish that APA-mediated 3′UTR shortening modulates myomiR repression to shape gene expression for robust myogenesis.

# Results

## Preferential 3′UTR shortening during satellite cell (SC) differentiation

To elucidate APA regulation during myogenic differentiation in a physiological context, we isolated SCs from adult mouse skeletal muscle, and induced their proliferation and differentiation into myotubes (MTs). Using QuantSeq (Moll et al, 2014), a method for high-resolution mapping of 3′ termini in polyadenylated RNAs, we profiled 3′UTR-APA changes in proliferating SCs versus differentiated MTs with two replicates (Fig. 1A,B).

Analysis of QuantSeq data revealed 2388 significant and reproducible APA events during SC differentiation (Fig. 1B). Intriguingly, in contrast to the well-documented association of 3′UTR lengthening with various cellular differentiation processes (Hoque et al, 2013; Gallicchio et al, 2023; Yang et al, 2022; Kang et al, 2023a), we observed a slight bias toward 3′UTR shortening (1264) over lengthening (1124) in MTs as compared to SCs (Fig. 1B; Dataset EV1). We were particularly interested in 3′UTR shortening as they deviate from the canonical APA regulation trend. KEGG enrichment analysis revealed that genes with shortened 3′UTRs were more enriched in pathways closely associated with muscle differentiation and function, compared to those with lengthened or unchanged 3′UTRs (comparing the number of terms, gene counts and adjusted *p*-values (Figs. 1C and EV1A) (Kitajima et al, 2020; Xia et al, 2021; Afroze and Kumar, 2019; Bohnert et al, 2018; Leduc-Gaudet et al, 2021; Yoon, 2017; Sanchez et al, 2013; Bloise et al, 2018), suggesting a potential functional role of 3′UTR shortening in myogenic progression.

While 3′UTR shortening is primarily attributed to increased proximal polyadenylation site (pA) usage in the nucleus, alternative mechanisms, such as selective degradation of long 3′UTR isoforms, might also contribute. If degradation was a major factor, we would expect reduced RNA levels in groups with shortened 3′UTR relative to those with unchanged or lengthened ones. However, transcript abundance of 3′UTR shortened group exhibited no such reduction (Fig. 1D), arguing against degradation as a dominant mechanism. To further investigate, we performed polyA RNA-seq on nuclear and cytoplasmic fractions isolated from SCs and MTs (Fig. 1E). Strikingly, differentiation led to an apparent shift toward proximal pA site usage in both the nucleus and the cytoplasm (Figs. 1F and EV1B). Specifically, 799 genes exhibited mRNA 3′UTR shortening compared to 412 with lengthening in the nucleus, while

in the cytoplasm, the corresponding numbers were 1531 and 303 (Figs. 1F and EV1B; Dataset EV2). The more pronounced shortening in the cytoplasmic fraction relative to nucleus might in part stem from preferential nuclear export of short isoform and/ or cytoplasmic degradation of long isoform (Neve et al, 2016; Chen et al, 2019; Tang et al, 2022; Bartel, 2018). The more extensive 3′UTR shortening detected in polyA RNA-seq compared to QuantSeq data may reflect differences in library preparation and analytical methods. Nevertheless, these data together indicate that 3′UTRs are preferentially shortened during SC differentiation, predominantly mediated by APA regulation.

## Inhibition of 3′UTR shortening impairs myogenic differentiation

We sought to investigate whether 3′UTR shortening during SC differentiation is functionally important. Among genes exhibiting mRNA 3′UTR shortening, *Matr3* emerged as a key candidate. First, *Matr3* displayed pronounced 3′UTR shortening during myogenic differentiation (ranking in the top 1% among 1264 shortened events) (Fig. 2A; Dataset EV1). Second, its short 3′UTR isoform is specifically detected in skeletal and cardiac muscle tissues (Quintero-Rivera et al, 2015). Third, a patient with a truncated *MATR3* 3′UTR exhibited severe cardiac defects and impaired fine motor skills (Quintero-Rivera et al, 2015). Isoform-specific RT-qPCRs validated an increased short/long (S/L) ratio during differentiation of both SCs and C2C12 myoblasts (MBs) (Fig. EV2A). Northern blot further demonstrated elevated short and reduced long 3′UTR isoform levels in this process (Fig. 2B).

To assess the function of *Matr3* 3′UTR shortening, we blocked the usage of its proximal pA site in SCs using an antisense morpholino oligonucleotide (*Matr3*-pAMO), followed by differentiation induction (Figs. 2C and EV2B). *Matr3*-pAMO treatment efficiently reduced the S/L ratio, a result of decreased short isoform and elevated long isoform levels (Figs. 2D and EV2C). Significantly, SC differentiation was apparently impaired upon blockage of *Matr3* proximal pA site, as evidenced by fewer multinucleated (>20) MTs and an increased population of cells with fewer nuclei (≤5) (Fig. 2E,F). Consistent with this morphological defect, the expression of *Myh1* and *Myh3*, two terminal differentiation markers (Bryson-Richardson and Currie, 2008; Allen et al, 2001), were both reduced in MTs with *Matr3*-pAMO treatment as compared to control (Cntl) AMO treatment (Fig. 2G). These data indicate that 3′UTR shortening of *Matr3* is important for efficient myogenic differentiation.

To explore whether this regulatory importance might extend to other genes, we selected six other genes (*Cpd*, *Foxp1*, *Il6st*, *Neo1*, *RalA*, and *Sec63*) that exhibited significant 3′UTR shortening and/ or known roles in muscle function (Fig. EV2D; Dataset EV1) (Nozaki et al, 2012; Fix et al, 2018; Bae et al, 2009; Neyroud et al, 2021). We used steric-blocking antisense oligonucleotides (ASOs) to inhibit their proximal pA site usage in MBs prior to differentiation induction (Fig. EV2E). No canonical poly(A) signal (PAS) hexamer or variant was found nearby the *Foxp1* proximal cleavage site, we therefore used an ASO to target an A/T-rich sequence upstream of the cleavage site (Fig. EV2E). ASO treatment apparently reduced S/L ratio for all target genes (Fig. EV2F). Significantly, inhibition of proximal pA site in *Foxp1*, *Neo1*, *RalA*, and *Sec63*, but not that of *Cpd* and *Il6st*, reproducibly impaired MT formation, as demonstrated by reduced myotube length (Fig. 2H–K)

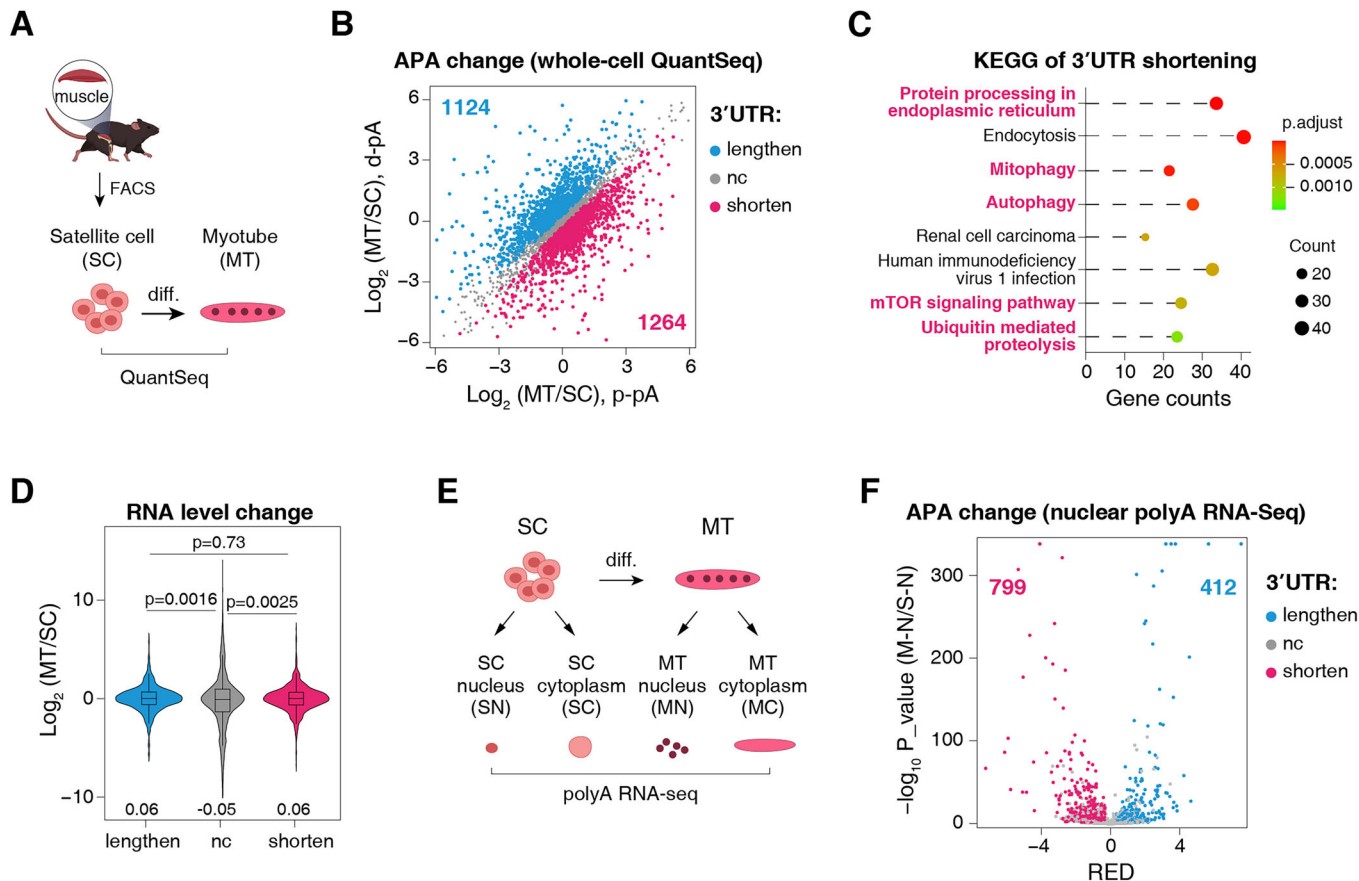

**Figure 1. Preferential 3′UTR shortening during satellite cell (SC) differentiation.**

(A) A workflow to identify the 3′ end of polyadenylated RNAs in proliferating SCs and their corresponding differentiated MTs using QuantSeq. diff, differentiation. (B) APA changes identified by QuantSeq from two biological replicates in MTs versus SCs. Genes exhibiting significant mRNA 3′UTR lengthening (blue) or shortening (magenta) are indicated. Gene counts are shown. p-pA, proximal polyadenylation site; d-pA, distal polyadenylation site. (C) KEGG pathway enrichment analysis of genes exhibiting mRNA 3′UTR shortening during myogenic differentiation. *P*-values were calculated using a hypergeometric test and adjusted for multiple hypothesis testing via the Benjamini-Hochberg method. (D) Violin plot illustrating RNA expression level changes in MTs versus SCs based on QuantSeq data analysis. Genes are grouped based on 3′ UTR changes (lengthened, unchanged (nc), and shortened). Lengthen, *n* = 1124; nc, *n* = 15135; shorten, *n* = 1264. *n*: the number of differential APA change genes. The central line of the boxplot signifies the median value (shown in the plot), the box borders represent the first (Q1) and third (Q3) quartiles, and the whiskers extend from the box to the extreme values within 1.5 times the inter quartile range (from Q1 to Q3). Statistical significance was assessed using the two-sided Wilcoxon rank-sum test (*P*-values shown). (E) A workflow for fractionated polyA RNA-seq in SCs and MTs. SN, SC nucleus; SC, SC cytoplasm; MN, myotube nucleus; MC, myotube cytoplasm. (F) Volcano plot showing the APA changes in MN versus SN. APA changes were quantified using RED (relative expression difference, Δlog₂[d-pA/p-pA]) (Wang and Tian, 2020). Statistically significant APA events were identified using a false discovery rate (FDR)-adjusted *p*-value < 0.05 (Fisher's exact test). Genes exhibiting significant mRNA 3′UTR shortening and lengthening are indicated in magenta and blue, respectively. Gene counts are shown.

and downregulated *Myh1* and *Myh3* expression compared to control (Cntl) ASO-treated cells (Fig. 2L). Together, these data indicate that 3′UTR shortening of at least a subset of genes is critical for efficient myogenic differentiation.

## CFI downregulation makes important contribution to 3′ UTR shortening during myogenic differentiation

How are 3′UTRs shortened during myogenic differentiation? Differential expression of 3′ processing factors is one of the major mechanisms for APA regulation (Tian and Manley, 2017; Lackford et al, 2014; Li et al, 2015; Martin et al, 2012; Takagaki et al, 1996). Consistent with this view, we found that 3′ processing factors were overall downregulated in MTs compared to SCs (Fig. 3A). To assess whether this downregulation could drive 3′UTR shortening, we

analyzed published 3′ Region Extraction And Deep Sequencing (3′ READS) data in MBs following knockdown (KD) of various 3′ processing factors, specifically focusing on factors whose KD induced significant 3′UTR shortening (log₂[L/S] < −0.8) (Li et al, 2015). Among these, components of CFI emerged as the most prominent regulators, with CFI-68 and CFI-25 KD recapitulated 67% and 64% of differentiation-induced shortened events, respectively (Fig. 3B). This strong phenotypic overlap suggests that downregulation of CFI components serve as a major driver of 3′ UTR shortening during myogenesis.

CFI facilitates the usage of distal pA site through binding to UGUA motifs (Zhu et al, 2018; Yang et al, 2010, 2011; Martin et al, 2012; Brown and Gilmartin, 2003). In support of its role in regulating 3′UTR shortening during myogenic differentiation, we found that UGUA motif were enriched in the alternative UTRs (aUTRs) of RNAs

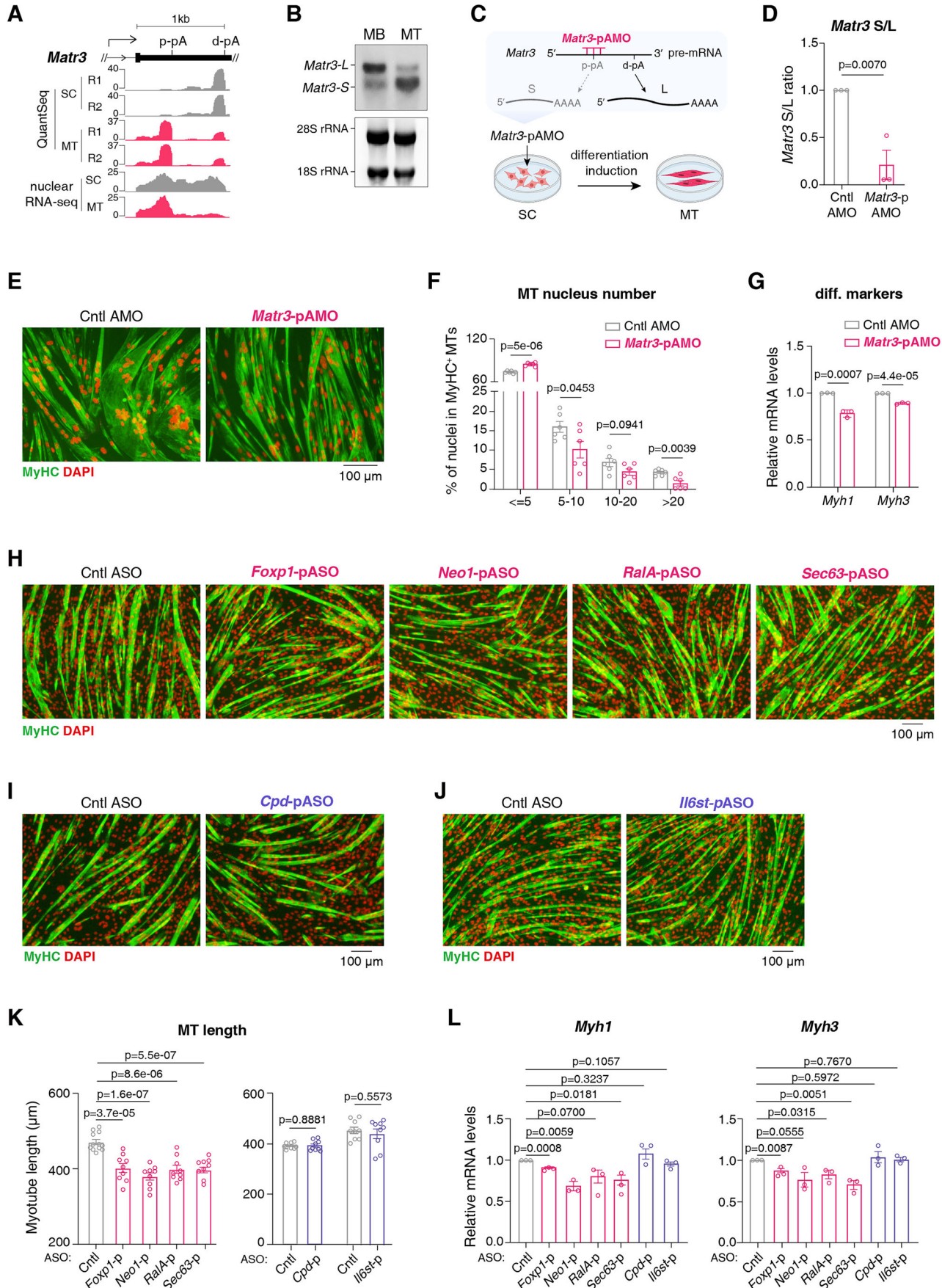

**Figure 2. Inhibition of 3′UTR shortening impairs myogenic differentiation.**

(**A**) Screenshot of QuantSeq and nuclear RNA-seq signals of *Matr3* 3′UTR. (**B**) Northern blot to detect the levels of *Matr3* short and long 3′UTR isoforms in C2C12 myoblasts (MB) and myotubes (MT). 28S and 18S rRNAs were used as loading controls. (**C**) Schematic illustration of the strategy for inhibiting *Matr3* proximal pA site usage using an AMO. (**D**) Isoform-specific RT-qPCR to measure *Matr3* S/L (short/long) isoform ratios in MTs differentiated for 2 days from SCs treated with either control AMO (Cntl AMO) or *Matr3*-pAMO ($n = 3$). Error bars, mean ± SEM. *P*-values, two-sided unpaired student's t test. (**E**) Representative immunofluorescence staining images of MyHC (green) and DAPI (red) in MTs differentiated from SCs treated with Cntl AMO or *Matr3*-pAMO. Scale bar, 100 μm. (**F**) Quantification of myotube formation efficiency in SCs treated with Cntl AMO or *Matr3*-pAMO. MyHC+ myotubes were categorized by nuclear numbers. Data were obtained from six images across three biological replicates. Error bars, mean ± SEM. *P*-values, two-sided unpaired student's t test. (**G**) RT-qPCR to measure the mRNA levels of *Myh1* and *Myh3* in MTs differentiated from SCs treated with Cntl AMO or *Matr3*-pAMO ($n = 3$). Bars represent RNA abundance normalized to *Gapdh*. Error bars, mean ± SEM. *P*-values, two-sided unpaired student's t test. (**H**) Representative immunofluorescence staining images of MyHC (green) and DAPI (red) in MTs differentiated from MBs treated with the Cntl, *Foxp1*-, *Neo1*-, *RalA*- and *Sec63*-*p*ASO, respectively. Scale bar, 100 μm. (**I**) Representative immunofluorescence staining images of MyHC (green) and DAPI (red) in MTs differentiated from MBs treated with Cntl ASO or *Cpd*-pASO. Scale bar, 100 μm. (**J**) Representative immunofluorescence staining images of MyHC (green) and DAPI (red) in MTs differentiated from MBs treated with Cntl ASO or *Il6st*-pASO. Scale bar, 100 μm. (**K**) Quantification of tube length of MTs differentiated from MBs treated with different ASOs. Data were obtained from nine images across three biological replicates, with the exception of the Cntl ASO group (left), for which data were obtained from twelve images across four biological replicates. Error bars, mean ± SEM. *P*-values, two-sided unpaired student's t test. (**L**) RT-qPCR to measure the mRNA levels of *Myh1* and *Myh3* in MTs differentiated from MBs treated with different ASOs ($n = 3$). Bars represent RNA abundance normalized to *Gapdh*. Error bars, mean ± SEM. *P*-values, two-sided unpaired student's t test. Source data are available online for this figure.

undergoing 3′UTR shortening than in those showing lengthening and no change. This enrichment was particularly pronounced for RNAs that shortened upon CFI-68 knockdown (Fig. 3C). Furthermore, analysis of published CFI-68 tRIP-seq data from C2C12 myoblasts (Kawachi et al, 2021) revealed stronger CFI-68 binding on 3′UTR shortened RNAs that were responsive to both differentiation induction and CFI-68 knock-down (Fig. 3D).

qPCR and western blot analyses confirmed downregulation of CFI-25 and CFI-68 upon myogenic differentiation (Fig. 3E,F). To further examine the role of CFI in 3′UTR shortening, we knocked down CFI-68 in MBs and assessed *Matr3* APA change. For comparison, we also depleted PABPN1 and PABPC1, which also influence a significant subset of differentiation-induced shortening events (Fig. 3B). CFI-68 KD significantly increased *Matr3* S/L ratio, mimicking the effect of myogenic differentiation, whereas PABPN1 and PABPC1 depletion had no such impact (Fig. 3G,H). Consistent with CFI's regulatory role, we identified two conserved UGUA motifs upstream of *Matr3* distal pA site (Fig. 3I). To test whether CFI regulates *Matr3* APA more directly, we inserted *Matr3* 3′UTR downstream of the luciferase coding sequence, with UGUA motifs intact (WT-Rz) or deleted (ΔUGUA-Rz), followed by a ribozyme (Rz) to block plasmid-derived polyadenylation (Fig. 3J). When expressed in MBs, ΔUGUA-Rz exhibited significantly higher S/L ratio compared to WT-Rz (Fig. 3J), demonstrating that CFI binding promotes distal pA site usage in *Matr3*. Collectively, these results establish that CFI downregulation during myogenic differentiation led to proximal pA site selection, resulting in preferential 3′UTR shortening.

## 3′UTR shortening positively impacts polysome engagement during myogenic differentiation

We next asked how 3′UTR shortening promotes myogenic differentiation. 3′UTR is known to influence various aspects of RNA metabolism, including mRNA localization, stability, and translation (Tian and Manley, 2017; An et al, 2008; Lau et al, 2010; Berkovits and Mayr, 2015; Spies et al, 2013). During SC differentiation, RNAs with shortened 3′UTRs exhibited higher expression levels compared to those with unchanged 3′UTRs, while showing similar expression levels to transcripts with lengthened 3′UTRs (Fig. 1D), suggesting that 3′UTR length alterations do not universally influence RNA stability. To assess the global impact of

3′UTR changes on nuclear RNA export, we analyzed nuclear and cytoplasmic polyA RNA-seq data from SCs and MTs (Fig. 1E), but observed no significant difference in cytoplasm-to-nucleus (C/N) ratio between RNAs with 3′UTR shortened, unchanged, or lengthened (Fig. 4A), indicating that 3′UTR shortening does not broadly affect RNA export during SC differentiation.

To obtain clues on whether 3′UTR shortening affects translation during myogenic differentiation, we performed polysome profiling to assess polysome engagement in MBs and MTs (Fig. 4B). Notably, mRNAs with lengthened or unchanged 3′UTRs showed significant decreases in polysome association in MTs versus MBs (Fig. 4C), suggestive of their overall translation repression during myogenic differentiation. In contrast, mRNAs with shortened 3′UTR exhibited no such decrease in polysome engagement (Fig. 4C), suggesting that 3′UTR shortening helps escape from this repression. Consistent with this, mRNAs with greater 3′UTR shortening tended to exhibit higher polysome association in MTs versus MBs (Fig. 4D). Together, these findings suggest that 3′UTR shortening buffer mRNAs from overall translational repression during myogenic differentiation.

## 3′UTR shortening as a mechanism to evade targeting by myomiRs

During myogenic differentiation, the dramatic induction of muscle-specific miRNAs (myomiRs), key regulators that repress proliferative factors (Boutet et al, 2012; De Morree et al, 2019; Chen et al, 2010), creates a need for escape mechanisms for differentiation-related genes. We hypothesized that 3′UTR shortening may provide such a mechanism for ensuring the expression of these genes.

To test this hypothesis, we analyzed public AGO2 CLIP data from MBs and MTs (Zhang et al, 2014), focusing on AGO2 binding changes in aUTRs of RNAs classified by 3′UTR changes (shortened, unchanged, and lengthened) (Fig. 4E). To ensure our analysis specifically reflected APA regulation rather than cytoplasmic mRNA decay, we restricted our analysis to RNAs with consistent 3′UTR changes in whole-cell QuantSeq and nuclear polyA RNA-seq datasets. Among RNAs with shortened 3′UTRs, 68% exhibited significantly increased AGO2 binding in their aUTRs following differentiation, compared to 53% and 55% of RNAs with unchanged or lengthened 3′UTRs (Fig. 4F; Dataset EV3), suggesting that miRNA targeted RNAs preferentially undergo 3′UTR

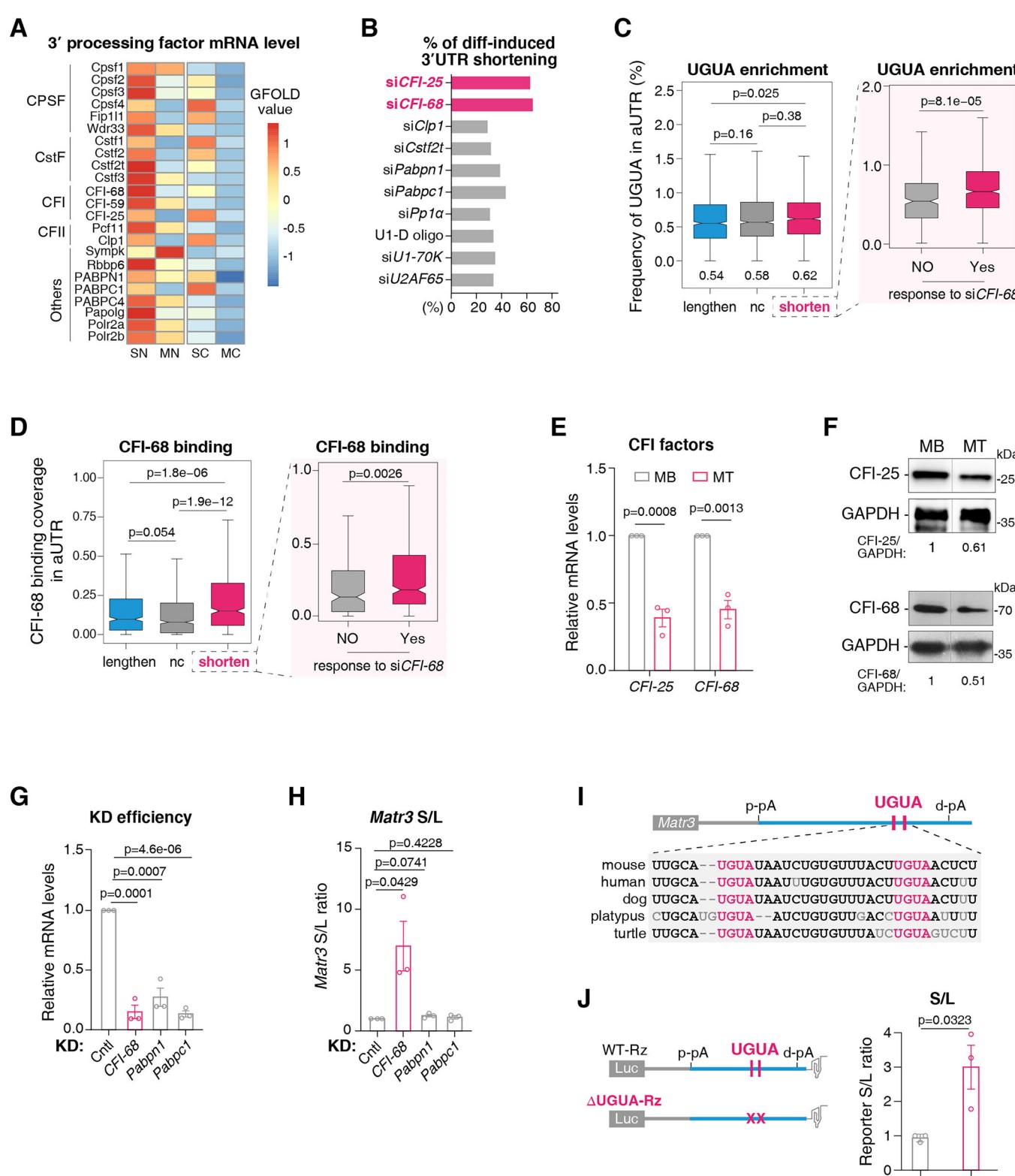

**Figure 3. CFI downregulation makes important contribution to 3′UTR shortening during muscle differentiation.**

(A) Heatmap exhibiting expression levels of 3′ processing factors in the nucleus and the cytoplasm of MTs and SCs, based on polyA RNA-seq data. Differentially expressed genes were ranked based on the GFOLD values. (B) Bar plot showing the percentage of differentiation-induced 3′UTR shortening events that also exhibit significant shortening upon depletion of the indicated 3′ processing factors in MBs. Analysis of the latter is based on published 3′ READS data (Li et al, 2015). (C) Boxplot comparing the frequency of UGUA motifs in the alternative UTR (aUTR) regions across different gene sets categorized by 3′UTR length change (left) and response to CFI-68 downregulation (siCFI-68) (right). Frequency is calculated as the number of UGUA motifs normalized to aUTR length. Lengthen, $n = 1124$; nc, $n = 2000$ (random); shorten, $n = 1264$; NO (response to siCFI-68), $n = 399$; Yes (response to siCFI-68), $n = 865$. Statistical significance was assessed using the two-sided unpaired student's t test (P-values shown). (D) Boxplot comparing CFI-68 binding coverage on mRNA aUTRs across gene sets categorized by 3′UTR length change (left) and response to CFI-68 downregulation (siCFI-68) (right). Lengthen, $n = 531$; nc, $n = 766$; shorten, $n = 644$; NO (response to siCFI-68), $n = 230$; Yes (response to siCFI-68), $n = 414$. Statistical significance was assessed using the two-sided Wilcoxon rank-sum test (P-values shown). Analysis of CFI-68 binding is based on published tRIP-seq data (Kawachi et al, 2021). (E) RT-qPCR to detect mRNA levels of CFI-25 and CFI-68 in MBs and MTs ($n = 3$). Bars represent RNA abundance normalized to Gapdh. Error bars, mean ± SEM. P-values, two-sided unpaired student's t test. (F) Western blot to detect CFI-25 and CFI-68 protein levels in MBs and MTs. Relative protein abundance to GAPDH is shown at the bottom. The gray line indicates the boundary where irrelevant lanes from the same blot were cropped. (G) RT-qPCR to measure knockdown (KD) efficiencies of CFI-68, Pabpn1, and Pabpc1 ($n = 3$). Bars represent RNA abundance normalized to Gapdh. Error bars, mean ± SEM. P-values, two-sided unpaired student's t test. (H) Isoform-specific RT-qPCR to examine Matr3 S/L ratios in KD samples ($n = 3$). Error bars, mean ± SEM. P-values, two-sided unpaired student's t test. (I) Schematic illustration of two conserved UGUA motifs upstream of Matr3 distal pA site. (J) Schematic illustration of reporter plasmids (left). Isoform-specific RT-qPCR to detect the reporter S/L ratio in MBs transfected with the indicated reporter plasmids (right) ($n = 3$). Error bars, mean ± SEM. P-values, two-sided unpaired student's t test. For boxplots in (C) and (D), the central line of the boxplot signifies the median value, the box borders represent the first (Q1) and third (Q3) quartiles, and the whiskers extend from the box to the extreme values within 1.5 times the interquartile range (from Q1 to Q3). Source data are available online for this figure.

shortening. This possibility was further supported by myomiR target site analysis (Fig. 4E; see Methods), which demonstrated that RNAs with shortened 3′UTRs were more likely to contain myomiR target sites within increased AGO2 binding peaks in aUTRs (28%) versus lengthened (16%) or unchanged (14%) 3′UTRs (Fig. 4G; Dataset EV3). Notably, three (Matr3, Foxp1, and Sec63) out of five RNAs whose 3′UTR shortening was functionally required for differentiation contained myomiR target sites (Figs. 2E–H,K,L and 4H; Appendix Fig. S1A; Dataset EV3). In contrast, neither of the two RNAs whose shortening had no impact on differentiation harbored myomiR target sites (Fig. 2I–L; Dataset EV3).

RNAs exhibiting differentiation-induced 3′UTR shortening and increased AGO2 binding showed moderate conservation (0.7–0.8) of myomiR target sites within aUTRs, polyadenylation signals (PASs) in proximal and distal pA sites (Fig. 4I). Notably, both proximal and distal PASs were significantly more conserved in shortened genes compared to non-shortened ones (Fig. 4J) and in myomiR target site containing genes versus lacking ones (Fig. 4K). These evolutionary patterns indicate that 3′UTR shortening represents a conserved mechanism to counteract myomiR repression during myogenic differentiation, enabling precise regulation of myogenic gene expression.

## The APA-miRNA interplay fine-tunes *Matr3* expression for efficient muscle differentiation

Our data so far indicate that during SC differentiation, the expression of many genes is regulated by two mechanisms: APA-mediated 3′UTR shortening and miRNA-dependent repression. To dissect the interplay between these two mechanisms, we focused on *Matr3* as a model gene. *Matr3*-pAMO treatment, which blocks 3′UTR shortening and impairs myotube formation (Fig. 2D–G), reproducibly reduced *Matr3* protein levels (Fig. 5A), indicating that 3′UTR shortening ensures sustained *Matr3* expression required for SC differentiation. This conclusion was further reinforced by *Matr3* KD experiments, which similarly disrupted myogenic differentiation (Fig. 5B–D; Appendix Fig. S2A) (Banerjee et al, 2017).

Our sequence analysis identified conserved target sequence for miR-1/206 within AGO2-binding peaks in the aUTR of *Matr3* (Fig. 4H; Appendix Fig. S2B; Dataset EV3). We used luciferase

reporter assays to validate its direct targeting by miR-1/206. In HEK293 cells (which lack endogenous miR-1/206), luciferase activity of construct containing wild-type *Matr3* 3′UTR fragment (Luc-WT) was significantly reduced by co-expressed miR-1 or miR-206, while the Luc-Mut harboring mutated target site remained unaffected (Fig. 5E,F). Consistent with induction of miR-1/206 upon differentiation (Kim et al, 2006; Chen et al, 2010), Luc-WT showed significantly reduced luciferase activity in MTs versus MBs, whereas Luc-Mut displayed no such reduction (Fig. 5E,G).

Interestingly, miR-1/206-mediated *Matr3* repression also appears functionally important in myogenic differentiation, as blocking their access to *Matr3* mRNA using a target-site-specific AMO increased *Matr3* protein abundance and impaired differentiation (Fig. 5H–K). In agreement with this, *Matr3* over-expression also led to impaired myogenic differentiation (Fig. 5L–N; Appendix Fig. S2C). Notably, these results align with clinical observations of MATR3 upregulation in a patient with truncation of distal region of *MATR3* 3′UTR (Quintero-Rivera et al, 2015). Together, these data reveal a sophisticated balance where APA and miRNA pathways antagonistically regulate the expression of *Matr3* for proper myogenic differentiation.

## Genetically mutating *Matr3* proximal pA site in mice impairs muscle regeneration

To assess the biological significance of *Matr3* APA regulation in myogenesis, we generated a mouse model with proximal pA site of *Matr3* mutated (*Matr3*^mpA/mpA) (Fig. 6A; see Methods for details). Northern blot analysis confirmed successful disruption of *Matr3* proximal pA site usage, as demonstrated by the absence of the short isoform and concomitant accumulation of the long isoform in skeletal muscle (Fig. 6B). *Matr3*^mpA/mpA mice were born at expected Mendelian ratios and showed normal appearance and body weight (Fig. EV3A–C).

Consistent with miR-1/206-mediated repression, we observed significantly reduced *Matr3* protein level in skeletal muscle of *Matr3*^mpA/mpA mice compared to *Matr3*^+/+ littermates (Fig. 6C). Importantly, this reduction was tissue-specific, as *Matr3* expression remained unchanged in lung, where miR-1/206 expression are not expressed (Fig. 6C) (Kim et al, 2006). Despite these molecular

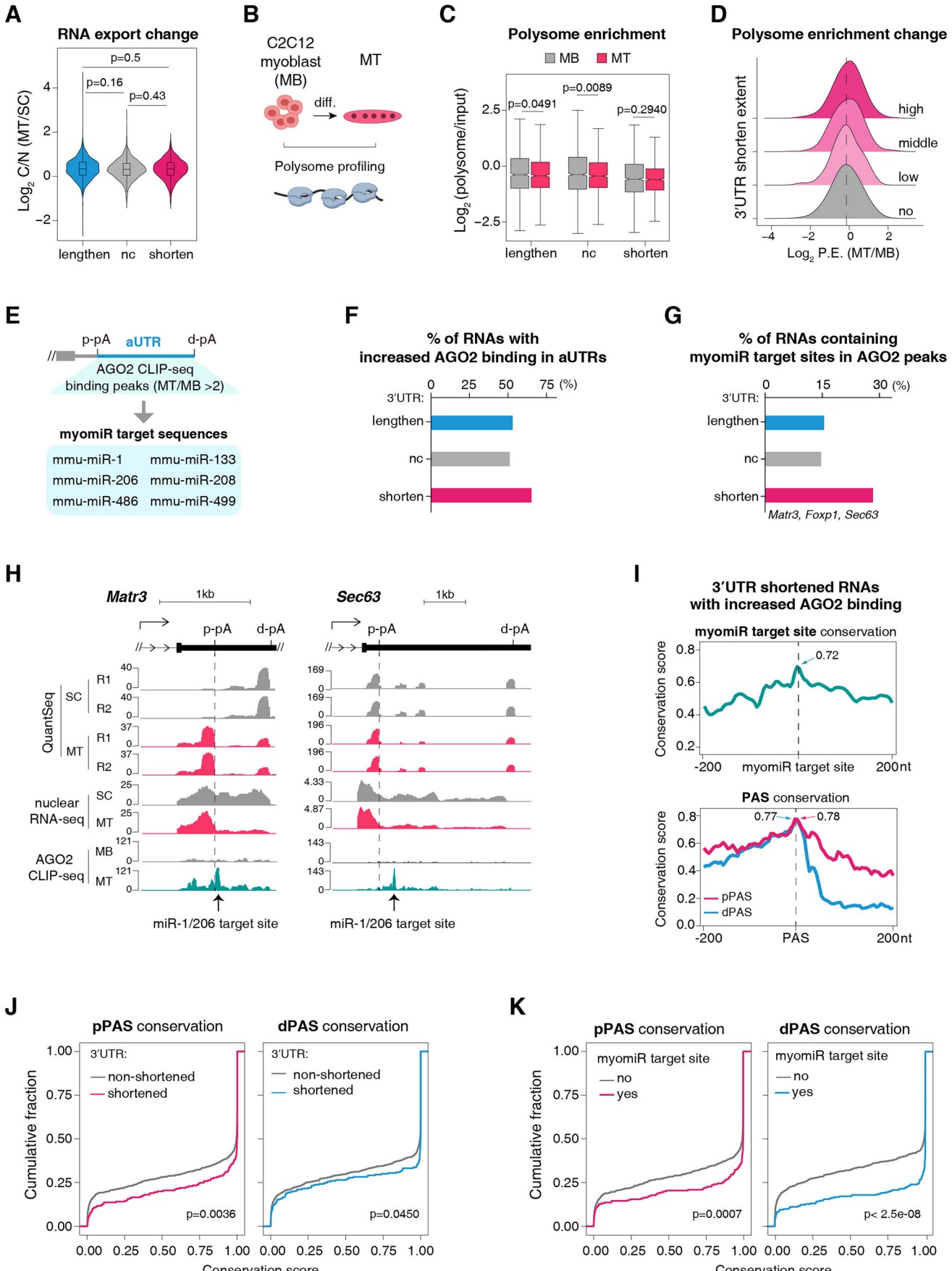

**Figure 4.** 3′UTR shortening serves as a mechanism to evade targeting by myomiRs.

(A) Violin plot illustrating RNA cytoplasm-to-nucleus (C/N) ratio changes in MTs versus SCs. Genes are classified into three groups based on 3′UTR length changes. Lengthen, $n = 1104$; nc, $n = 14330$; shorten, $n = 1236$. Statistical significance was assessed using the two-sided Wilcoxon rank-sum test (P-values shown). (B) Schematic illustration of polysome profiling in MBs and MTs. (C) Boxplots illustrating the polysome enrichment on mRNAs with different differentiation-induced 3′UTR length changes in MBs and MTs. Lengthen, $n = 1077$; nc, $n = 9672$; shorten, $n = 1211$. Statistical significance was assessed using the two-sided Wilcoxon rank-sum test (P-values shown). (D) The distribution of polysome enrichment (P.E.) changes across distinct 3′UTR shortening magnitude groups (high, middle, low, and no shortening). Genes with shortened 3′UTRs are equally divided into three groups. (E) A workflow for identifying increased AGO2 binding peaks in aUTRs and candidate myomiR target sequences within these peaks. (F) Bar plot showing the proportion of mRNAs with increased AGO2 binding peaks across distinct 3′UTR length change groups. (G) Bar plot showing the proportion of mRNAs with myomiR target sites across distinct 3′UTR length change groups. (H) Screenshot of QuantSeq, nuclear RNA-seq, and AGO2 CLIP-seq signals of *Matr3* and *Sec63* 3′UTR. Predicted miR-1/206 target sites are indicated by arrows. (I) Conservation scores of myomiR target sites and their flanking regions in mRNAs identified in (G) (top). Conservation scores of proximal (pPASs) or diatal (dPASs) polyadenylation signals and their respective flanking regions in mRNAs identified at (F) (bottom). The conservation scores of myomiR target sites, pPASs and dPASs are labeled in the plot. (J) Cumulative distribution curves comparing evolutionary conservation scores of pPAS (left) and dPAS (right) between mRNAs with shortened versus non-shortened 3′UTRs. Statistical significance was assessed using the two-sided Wilcoxon rank-sum test (P-values shown). (K) Cumulative distribution curves comparing evolutionary conservation scores of pPAS (left) and dPAS (right) between mRNAs with or without myomiR target sites. Statistical significance was assessed using the two-sided Wilcoxon rank-sum test (P-values shown). For boxplots in (A) and (C), the central line of the boxplot signifies the median value, the box borders represent the first (Q1) and third (Q3) quartiles, and the whiskers extend from the box to the extreme values within 1.5 times the interquartile range (from Q1 to Q3).

changes, adult *Matr3*[mpA/mpA] mice showed normal TA muscle mass, fiber cross-sectional area, and physical performance in grip strength, rotarod test, or treadmill test (Fig. EV3D–J), indicating that *Matr3* APA regulation is dispensable for baseline muscle development and function.

Given the established role of myogenic differentiation in muscle repair (Fu et al, 2022; Sousa-Victor et al, 2022), we next investigated whether *Matr3* 3′UTR shortening impacts muscle regeneration. Following single dose of cardiotoxin (CTX) treatment of TA muscles, *Matr3*[mpA/mpA] mice displayed moderately but reproducibly compromised regenerative potential compared to the *Matr3*[+/+] littermates, characterized by smaller myofiber areas (Fig. 6D,E). This regenerative deficit became more pronounced after successive injury challenges, with *Matr3*[mpA/mpA] mice showing significantly higher proportion of small-area fibers compared to littermate controls (Fig. 6F,G). These results show that *Matr3* 3′UTR shortening is important for myogenic differentiation in vivo.

To determine whether this impaired muscle regeneration in mutant mice stemmed from intrinsic SC defects, we isolated SCs from *Matr3*[mpA/mpA] and *Matr3*[+/+] mice. While EdU incorporation assays revealed normal proliferative capacity of *Matr3*[mpA/mpA] SCs (Fig. EV3K,L), differentiation assays demonstrated clear impairments, with *Matr3*[mpA/mpA] SCs forming fewer multinucleated myotubes and showing reduced expression of differentiation makers (Fig. 6H–J). These findings establish that *Matr3* 3′UTR shortening is essential for myogenesis during muscle regeneration.

### Dysregulated 3′UTR shortening in muscle/neuromuscular diseases

We wonder whether there is a potential relevance of 3′UTR shortening in muscle/neuromuscular disorders. To explore this, we analyzed APA changes using public transcriptome data from Duchenne muscular dystrophy (DMD) and spinal muscular atrophy (SMA) mouse models, and myotonic dystrophy (DM) human patient samples (Georgieva et al, 2022; Batra et al, 2017; Meijboom et al, 2021).

A subset of differentiation-induced 3′UTR shortened genes exhibited dysregulated 3′UTR patterns in diseased skeletal muscle (Fig. 7A). Notably, significantly more genes displayed RNA 3′UTR lengthening than shortening across all disease conditions examined (Fig. 7A), supporting the potential contribution of APA regulation

in muscle biology. Gene Ontology (GO) analysis of affected genes highlighted strong enrichment of muscle development-related terms, especially in DMD (Figs. 7B and EV4A,B). While these diseases are primarily caused by mutations or repeat expansions of specific genes, such as dystrophin (Bez Batti Angulski et al, 2023), the pathological microenvironment may induce dysregulated 3′UTR shortening, which could possibly influence disease progression by modulating the expression of genes critical for muscle function.

Further analysis identified several mRNAs with consistent 3′UTR lengthening across different disease datasets (Fig. 7C; Dataset EV4). Among these, *Wdr5*, a gene implicated in facioscapulohumeral muscular dystrophy (Mocciaro et al, 2023), showed consistent 3′UTR lengthening across DMD and DM samples (Fig. 7D; Dataset EV4), implying that its aberrant APA regulation may contribute to dystrophic phenotypes. Taken together, our results establish 3′UTR shortening as a crucial regulatory mechanism in myogenesis and suggest its disruption may contribute to the pathogenesis of muscle/neuromuscular disorders.

## Discussion

APA and miRNA represent two fundamental mechanisms of post-transcriptional regulation (Di Giammartino et al, 2011; Shi, 2012; Tian and Manley, 2017; Kuppusamy et al, 2013; Fabian and Sonenberg, 2012). While these pathways have been extensively studied independently, their interplay in critical biological processes remains poorly understood. Our study here reveals a critical regulatory paradigm in skeletal muscle differentiation, where 3′UTR shortening through APA serves to counterbalance myomiR induction, thereby fine-tuning gene expression programs essential for proper myogenesis.

Different than the general association of 3′UTR lengthening with cellular differentiation (Hoque et al, 2013; Gallicchio et al, 2023; Yang et al, 2022; Kang et al, 2023a), we observed preferential 3′UTR shortening during primary muscle stem cell differentiation. We propose this unique 3′UTR shortening dynamic in muscles reflects a compensatory mechanism to counteract the concurrent dramatic induction of myomiRs (e.g., miR-1, miR-206), which

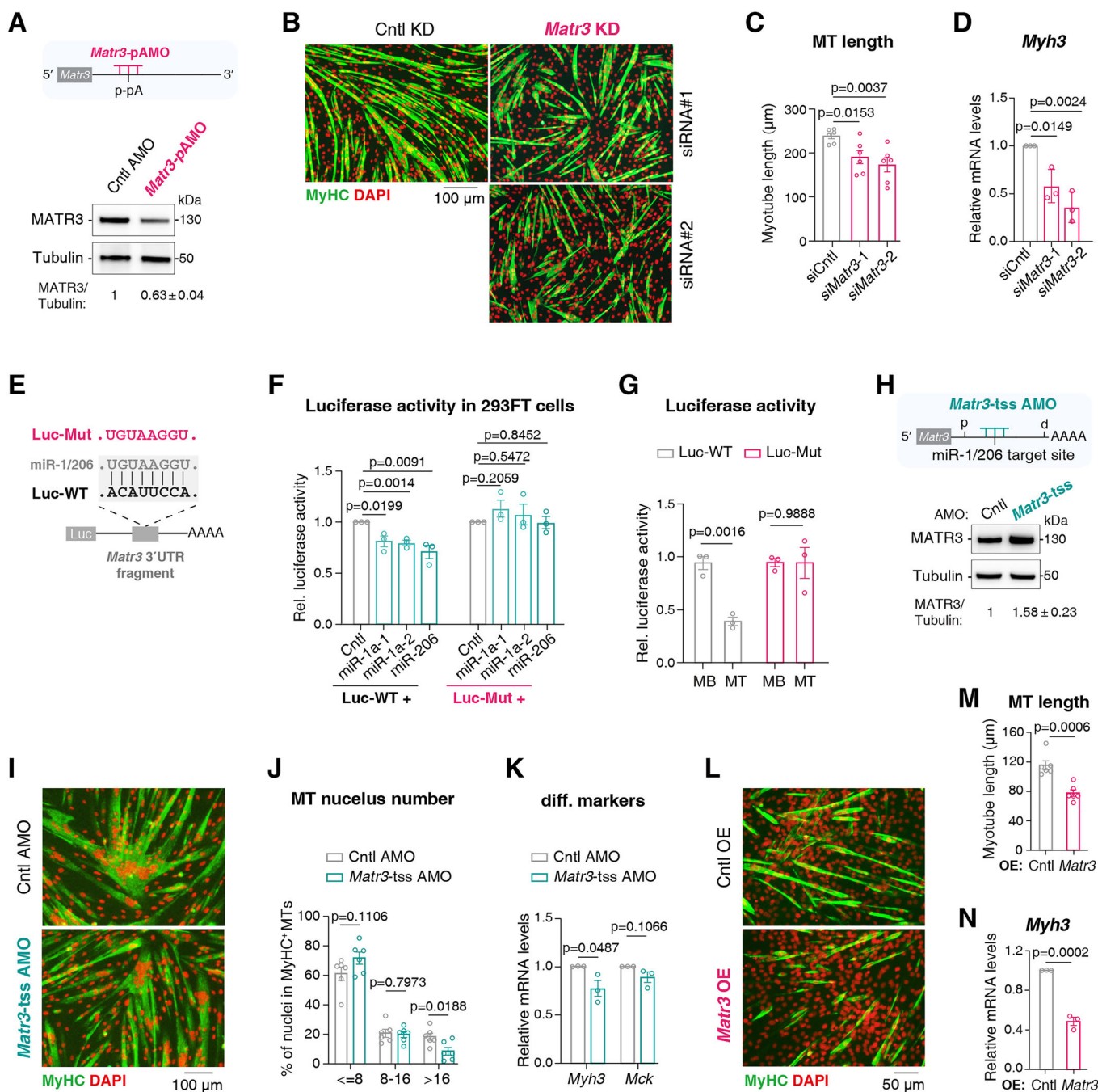

potently silence proliferation genes (Dey et al, 2011; Boutet et al, 2012; De Morree et al, 2019; Chen et al, 2010). By shortening 3′ UTR length of transcripts, differentiation-critical genes (e.g., *Matr3*) evade myomiR-mediated repression, ensuring their sustained expression during myogenesis. Given the evolutionary conservation of APA and miRNA pathways, such 3′UTR shortening may represent a broader paradigm for safeguarding differentiation-critical genes—particularly in contexts where differentiation coincides with rapid miRNA upregulation. We note that 3′UTR lengthening was previously observed in immortalized C2C12 myoblast differentiation (Hoque et al, 2013). This

discrepancy is likely attributable to fundamental differences between primary stem cells and cell line models (Boontheekul et al, 2007; Maley et al, 1995). Satellite cells differentiate more synchronously and efficiently than C2C12 cells, forming myotubes that are more mature and closely resemble mature myofibers in vivo (Langelaan et al, 2011; Knežić et al, 2022).

Mechanistically, myogenic differentiation-induced 3′UTR shortening primarily results from downregulation of CFI proteins, which normally promote distal pA site usage (Fig. 8). Reduced CFI expression during differentiation shifts the balance toward proximal pA site usage. However, the mechanism underlying CFI

**Figure 5. The APA-miRNA interplay fine-tunes *Matr3* expression for efficient muscle differentiation.**

(A) Schematic illustration of using an AMO to block the usage of *Matr3* p-pA site (top). Western blot to examine *Matr3* protein levels in MTs differentiated from SCs treated with Cntl AMO or *Matr3*-pAMO (bottom). Relative protein abundance of MATR3 to Tubulin (mean ± SEM, $n = 3$) are shown at the bottom. (B) Representative immunofluorescence staining images of MyHC (green) and DAPI (red) in MTs differentiated from MBs treated with Cntl or *Matr3* siRNAs. Scale bar, 100 μm. (C) Quantification of tube length of MTs differentiated from MBs treated with different siRNAs. Data were obtained from six images across three independent replicates. Error bars, mean ± SEM. *P*-values, two-sided unpaired student's t test. (D) RT-qPCR to examine *Myh3* mRNA level in MTs differentiated from MBs treated with Cntl or *Matr3* siRNAs ($n = 3$). Bars represent RNA abundance normalized to *Gapdh*. Error bars, mean ± SEM. *P*-values, two-sided unpaired student's t test. (E) Schematic of luciferase reporters containing a fragment of *Matr3* 3'UTR with wild-type (Luc-WT) or mutated miR-1/206 target site (Luc-Mut). (F) Firefly luciferase activity (normalized to Renilla) was measured 48 h after HEK293 cells were co-transfected with luciferase reporter constructs (E) and miRNA expression plasmids ($n = 3$). Error bars, mean ± SEM. *P*-values, two-sided unpaired student's t test. (G) Firefly luciferase activity (normalized to Renilla) was measured at 24 h after MBs were transfected with Luc-WT or Luc-Mut, and at day 2 after differentiation induction (MT) ($n = 3$). Error bars, mean ± SEM. *P*-values, two-sided unpaired student's t test. (H) Schematic illustration of blocking miR-1/206 targeting site in *Matr3* 3'UTR using target-site-specific AMO (*Matr3*-tss AMO) (top). p, proximal polyadenylation site; d, distal polyadenylation site. *Matr3* protein levels in MTs differentiated for 2 days from SCs treated with Cntl or *Matr3*-tss AMO were detected (bottom). Relative protein abundance of MATR3 to Tubulin (mean ± SEM, $n = 2$) are shown at the bottom. (I) Representative immunofluorescence staining images of MyHC (green) and DAPI (red) in MTs differentiated for 2 days from SCs treated with Cntl or *Matr3*-tss AMO. Scale bar, 100 μm. (J) Quantification of myotube formation efficiency in SCs treated with control AMO or *Matr3*-tss AMO. MyHC$^+$ myotubes were categorized by nuclear numbers. Data were obtained from six images across three independent replicates. Error bars, mean ± SEM. *P*-values, two-sided unpaired student's t test. (K) RT-qPCR to measure the mRNA levels of *Myh3* and *Mck* in MTs differentiated from SCs treated with Cntl or *Matr3*-tss AMO ($n = 3$). Bars represent RNA abundance normalized to *Gapdh*. Error bars, mean ± SEM. *P*-values, two-sided unpaired student's t test. (L) Representative immunofluorescence staining images of MyHC (green) and DAPI (red) in MTs differentiated for 3 days from MBs transfected with Cntl or *Matr3* overexpression plasmids. Scale bar, 50 μm. (M) Quantification of tube length of MTs differentiated from MBs transfected with Cntl or *Matr3* overexpression plasmids. Data were obtained from six images across three independent replicates. Error bars, mean ± SEM. *P*-values, two-sided unpaired student's t test. (N) RT-qPCR to measure the mRNA level of *Myh3* in MTs differentiated from MBs transfected with Cntl or *Matr3* overexpression plasmids ($n = 3$). Bars represent RNA abundance normalized to *Gapdh*. Error bars, mean ± SEM. *P*-values, two-sided unpaired student's t test. Source data are available online for this figure.

downregulation during myogenic differentiation remains unclear. Given the functional synergy required between APA and miRNA programs, it is plausible to hypothesize that myomiRs may directly or indirectly regulate CFI components, creating a feedforward loop that coordinates post-transcriptional control during myogenic differentiation. Furthermore, transcription regulation may contribute to reduced CFI expression. Additionally, polysome association of both CFI-25 and CFI-68 decreased upon differentiation (Appendix Fig. S3A), indicating possible contribution of translational regulation to reduced CFI protein levels during myogenesis.

Our investigation of *Matr3* provided compelling evidence for the functional significance of both APA-mediated 3'UTR shortening and miRNA-mediated repression (Fig. 8). By employing an AMO to block proximal pA site usage in SCs, we observed impaired differentiation capacity. More importantly, genetic disruption of the proximal pA site in mice led to defective muscle regeneration due to impaired SC differentiation. Similarly, we established the critical role of miR-1/206-mediated *Matr3* repression in myogenesis, as inhibition of its miRNA binding site disrupted SC differentiation, phenocopying the effects of *Matr3* overexpression.

Beyond *Matr3*, we identified many other genes whose mRNAs exhibited APA-mediated 3'UTR shortening and increased AGO2 binding on aUTRs during myogenic differentiation, suggesting an extensive post-transcriptional network fine-tuning the myogenic program. Disrupting mRNA 3'UTR shortening in several of these genes likewise impaired differentiation. Notably, a subset of differentiation-induced 3'UTR shortening events demonstrated lengthening in muscle/neuromuscular disorders, implicating APA dysregulation in disease pathogenesis. We speculate that APA-mediated 3'UTR length control likely represents a more general principle for balancing regulation by miRNA alongside other factors (e.g., RNA-binding proteins, RNA modifications) during rapid cellular transitions requiring tunable gene expression. It would be valuable for future research to explore the implication of dysregulated APA in disease progression and assess the therapeutic value of restoring normal APA dynamics in muscle disorders.

# Methods

### Reagents and tools table

| Reagent/Resource | Reference or Source | Identifier or Catalog Number |
|---|---|---|
| **Experimental models** | | |
| C2C12 | ATCC | N/A |
| HEK293FT | ATCC | N/A |
| **Recombinant DNA** | | |
| Plasmid: Lut-WT | This paper | N/A |
| Plasmid: Luc-Mut | This paper | N/A |
| Plasmid: miRNA control | This paper | N/A |
| Plasmid: miR-1a-1 | This paper | N/A |
| Plasmid: miR-1a-2 | This paper | N/A |
| Plasmid: miR-206 | This paper | N/A |
| Plasmid: WT-Rz | This paper | N/A |
| Plasmid: ΔUGUA-Rz | This paper | N/A |
| Plasmid: Flag-Matr3 | This paper | N/A |
| Plasmid: Flag-GST | This paper | N/A |
| **Antibodies** | | |
| Rabbit polyclonal Anti-CFI-25 | Proteintech | 10322-1-AP |
| Rabbit polyclonal Anti-CFI-68 | Abcam | ab99347 |
| Mouse monoclonal Anti-GAPDH | Proteintech | 60004-1-Ig |
| Rabbit monoclonal Anti-MATR3 | Abcam | ab151739 |
| Mouse monoclonal Anti-Tubulin | Proteintech | 66031-1-Ig |

| Reagent/Resource | Reference or Source | Identifier or Catalog Number |
|---|---|---|
| Rabbit polyclonal Anti-Laminin | Abcam | Ab11575 |
| Mouse monoclonal Anti-MYHC | Abcam | Ab37484 |
| Rat monoclonal FITC-anti-CD45 | Invitrogen | 11-0451-82 |
| Rat monoclonal FITC-anti-CD31 | Invitrogen | 11-0311-82 |
| Rat monoclonal FITC-anti-SCA1 | Invitrogen | 11-5981-85 |
| Rat monoclonal Bition-anti-CD106(VCAM1) | Biolegend | 105704 |
| **Oligonucleotides and other sequence-based reagents** | | |
| siRNAs used in this study see Appendix Table S1 | This paper | N/A |
| AMOs used in this study see Appendix Table S2 | This paper | N/A |
| ASOs used in this study see Appendix Table S3 | This paper | N/A |
| Primers used in this study see Appendix Table S4 | This paper | N/A |
| **Chemicals, Enzymes and other reagents** | | |
| Cdiotoxin (CTX) | Sigma | 217503 |
| Collagenase II | Worthington Biochemical | LS004176 |
| DMEM | Gibco | 12100-061 |
| Dispase II | Roche | 04942078001 |
| DAPI | Sigma-Aldrich | 10236276001 |
| Endo-porter (PEG) | Genetools | N/A |
| Ham's F-10 | Gibco | 11550-043 |
| horse serum | Cytiva | SH30074.03 |
| IL-1α | Novoprotein | C041 |
| IL-13 | Novoprotein | CX56 |
| IFN-γ | Novoprotein | CM41 |
| TNF-α | Novoprotein | CF09 |
| bFGF | Novoprotein | C779 |
| Lipofectamine RNAiMAX | Invitrogen | 13778150 |
| Lipofectamine 2000 | Invitrogen | 11668019 |
| Lipofectamine LTX and Plus | Invitrogen | 15338100 |
| M-MLV reverse transcriptase | Promega | M1705 |
| penicillin/streptomycin | Gibco | 15070-063 |
| RQ1 RNase-Free DNase | Promega | M6101 |
| SYBR qPCR SuperMix | Novoprotein | E096-01B |

| Reagent/Resource | Reference or Source | Identifier or Catalog Number |
|---|---|---|
| TRIzol | Invitrogen | 15596018 |
| **Software** | | |
| Cutadapt (v2.6) | Martin, 2011 https://journal.embnet.org/index.php/embnetjournal/article/view/200 | http://cutadapt.readthedocs.io/en/stable/guide.html |
| STAR (v2.7.10) | Dobin et al, 2013 https://academic.oup.com/bioinformatics/article/29/1/15/272537 | https://github.com/alexdobin/STAR |
| deepTools | Ramírez et al, 2016 https://doi.org/10.1093/nar/gkw257 | https://deeptools.readthedocs.io/en/develop/ |
| MAAPER | Li et al, 2021 https://doi.org/10.1186/s13059-021-02429-5 | https://github.com/Vivianstats/MAAPER |
| DESeq2 | Love et al, 2014 https://doi.org/10.1186/s13059-014-0550-8 | https://github.com/thelovelab/DESeq2 |
| APAlyzer | Wang and Tian, 2020 https://academic.oup.com/bioinformatics/article/36/12/3907/5823886 | https://bioconductor.org/packages/release/bioc/html/APAlyzer.html |
| Subread - featureCounts | Liao et al, 2013 https://academic.oup.com/nar/article/41/10/e108/1075719 | http://subread.sourceforge.net/ |
| MACS (v2.2.7.1) | Zhang et al, 2008 https://doi.org/10.1186/gb-2008-9-9-r137 | N/A |
| Bowtie | Langmead and Salzberg, 2012 https://www.nature.com/articles/nmeth.1923 | http://bowtie-bio.sourceforge.net/index.shtml |
| Phastcons | Siepel et al, 2005 http://genome.cshlp.org/content/15/8/1034 | N/A |
| **Other** | | |
| Dual-Luciferase Reporter Assay System | Promega | E1910 |
| Dig Northern Starter Kit | Roche | 12039672910 |
| Cell-Light EdU Apollo 567 In Vitro Kit | RIBOBIO | C10310-1 |
| HiScript II 1st Strand cDNA Synthesis Kit | Vazyme | R212-01 |
| Lexogen's QuantSeq 3' mRNA-Seq V2 Library Prep Kit FWD | Lexogen | N/A |
| KOD -Plus- Mutagenesis Kit | Taraka | SMK-101 |
| ClonExpress Ultra One Step Cloning Kit | Vazyme | C115-02 |
| VAHTS Universal V8 RNA-seq Library Prep Kit for Illumina | Vazyme | NR605 |

## Plasmids and antibodies

To construct Luc-WT plasmid, a part of *Matr3* 3′UTR sequence (2980 to 3469; NM_010771.7) including miR-1/206 binding site was cloned into pmirGLO Dual-Luciferase miRNA Target Expression Vector (Promega). Luc-Mut plasmid was generated by mutagenesis using the KOD-Plus-Mutagenesis Kit (TOYOBO, SMK-101). The expression plasmids of miR-1 and miR-206 were obtained by inserting the corresponding pre-miRNA sequence into pcDNA3.0 (Invitrogen). To construct the WT-Rz plasmids, we cloned the *Matr3* long 3′UTR sequence into the pmirGLO Dual-Luciferase miRNA Target Expression Vector (Promega) and then inserted the HDV ribozyme sequence to block plasmid-derived polyadenylation. The ΔUGUA-Rz plasmid was constructed by mutagenesis. To construct the expression plasmids of Flag-MATR3 and Flag-GST, the corresponding sequences were separately cloned into pHAGE-Flag-IRES-ZsGreen vector (Fan et al, 2024) using ClonExpress Ultra One Step Cloning Kit (Vazyme, C115-01).

Antibodies used in this study include mouse anti-MHC (Abcam, ab37484), rabbit anti-CFI-25 (Proteintech, 10322-1-AP), rabbit anti-CFI-68 (Abcam, ab99347), mouse anti-GAPDH (Proteintech, 60004-1-Ig), rabbit anti-MATR3 (Abcam, ab151739), mouse anti-Tubulin (Proteintech, 66031-1-Ig), rabbit anti-Laminin (Abcam, ab11575), rat FITC-anti-CD31 (Invitrogen, 11-5981-85), rat FITC-anti-CD45 (Invitrogen, 11-0451-82), rat FITC-anti-SCA1 (Invitrogen, 11-5981-85), rat Biotin-anti-VCAM1 (Biolegend, 105704).

## Cell culture, differentiation induction and transfection

Satellite cells (SCs) were cultured as described previously (Wang et al, 2020; Fu et al, 2015). SCs were cultured in collagen-coated dishes with Ham's F-10 supplemented with 20% FBS (Cytiva, SH30396.03), 5 ng/mL IL-1α (Novoprotein, C041), 5 ng/mL IL-13 (Novoprotein, CX56), 10 ng/mL IFN-γ (Novoprotein, CM41), 10 ng/mL TNF-α (Novoprotein, CF09), 2.5 ng/ml bFGF (Novoprotein, C779) and 1% penicillin/streptomycin. Human HEK293 cells and mouse C2C12 myoblasts sourced from the American Type Culture Collection (ATCC), were cultured in DMEM (Gibco, 12100-061) supplement with 10% FBS and 1% penicillin/streptomycin. Myogenic differentiation was induced by switching culture media to DMEM supplement with 2% horse serum and 1% penicillin/streptomycin. All cells are regularly tested for mycoplasma contamination.

SCs were transfected with antisense morpholino oligonucleotide (AMO) at a concentration of 30 μM with Endo-Porter (Gene Tools, LLC), according to the manufacturer's instructions. Differentiation was induced 12 h post transfection and samples were anlyzed after 48 h differentiation.

siRNA and antisense oligonucleotide (ASO) transfection were performed using Lipofectamine RNAiMAX (Invitrogen, 13778150) according to the manufacturer's protocol. To attain the best transfection efficiency in C2C12 myoblasts, siRNAs and ASOs were transfected again 24 h after the first transfection. Plasmids were transfected into HEK293 cells and C2C12 myoblasts using Lipofectamine 2000 (Invitrogen, 11668019) and Lipofectamine LTX and Plus reagent (Invitrogen, 15338100), respectively, according to the manufacturer's protocols.

Sequences of siRNAs, AMOs and ASOs are provided in Appendix Tables S1–S3.

## Animals

All animal experiments were performed according to the guidelines of the Animal Care and Use Committee at Shanghai Institute of Biochemistry and Cell Biology, Center for Excellence in Molecular Cell Science, Chinese Academy of Sciences. Animal experiment protocols approved by the Institutional Animal Care and Use Committee (approval number: SIBCB-NAF-15-001-s217-032). Mice were housed in a specific pathogen-free (SPF) environment.

The *Matr3*^mpA/mpA mouse was generated using CRISPR-Cas9 genome editing technology, introducing a precise substitution of the pPAS within the *Matr3* 3′UTR with a BamHI restriction site (GGATCC). To avoid potential compensatory usage of cryptic PASs, three consecutive (≥2) adenine residues within the 12-nucleotide region immediately downstream of the pPAS were mutated to thymine. Furthermore, based on PolyA_DB database (Wang et al, 2018) (https://exon.apps.wistar.org/PolyA_DB/v3/), a 26-nucleotide region upstream of the pPAS (from position −203 to −177 relative to the pPAS) was modified by substituting consecutive (≥2) adenines with thymines to eliminate potential alternative cleavage sites. Primers of genotyping are listed in Appendix Table S4. All mice were maintained on a C57BL/6J background and underwent six generations of backcrossing. Both male and female mice were included in the study.

## Satellite cell isolation

Satellite cell isolation was performed as previously described (Han and Hu, 2019) with minor modifications. In brief, after mice were euthanized, hindlimb skeletal muscles were harvested, minced, and then digested in 775 U/ml Collagenase II (Worthington Biochemical, LS004176) in Wash medium ((Ham's F-10 (Gibco, 11550-043) supplemented with 1% penicillin/streptomycin (Gibco, 15070-063) and 10% horse serum (Cytiva, SH30074.03)) at 37 °C water bath for 60 min with intermittent shaking. The digested mixture was washed once with Wash medium and centrifuged at 500 × g for 5 min, and the supernatant was discarded. A second digestion was then performed using 100 U/ml Collagenase II and 1.1 U/ml Dispase II (Roche, 04942078001) in Wash medium at 37 °C for 30 min with intermittent shaking. The twice-digested muscle mixture was homogenized by pipetting through a 20 G needle ten times, then washed once with Wash medium and centrifuged to remove the supernatant. The pellet was resuspended in Wash medium, and the suspension was sequentially filtered through a 70 μm filter followed by a 40 μm filter, then centrifuged to collect the final cell pellet. After red blood cell lysis, the cells were washed and stained with fluorescently labeled antibodies against CD31, CD45, SCA1, and VCAM1 for 40 min on ice. Satellite cells were isolated by FACS using the CD31⁻CD45⁻SCA1⁻VCAM1⁺ gating strategy.

## RNA isolation and RT-qPCR

Total RNAs were extracted with TRIzol (Invitrogen, 15596018). cDNA synthesis was performed using random primers and M-MLV reverse transcriptase (Promega, M1705) or HiScript II 1st Strand cDNA Synthesis Kit (Vazyme, R212-01). For isoform-specific RT-qPCR, cDNA was synthesized using an isoform-sepcific RT primer and M-MLV reverse transcriptase. Quantitative PCR

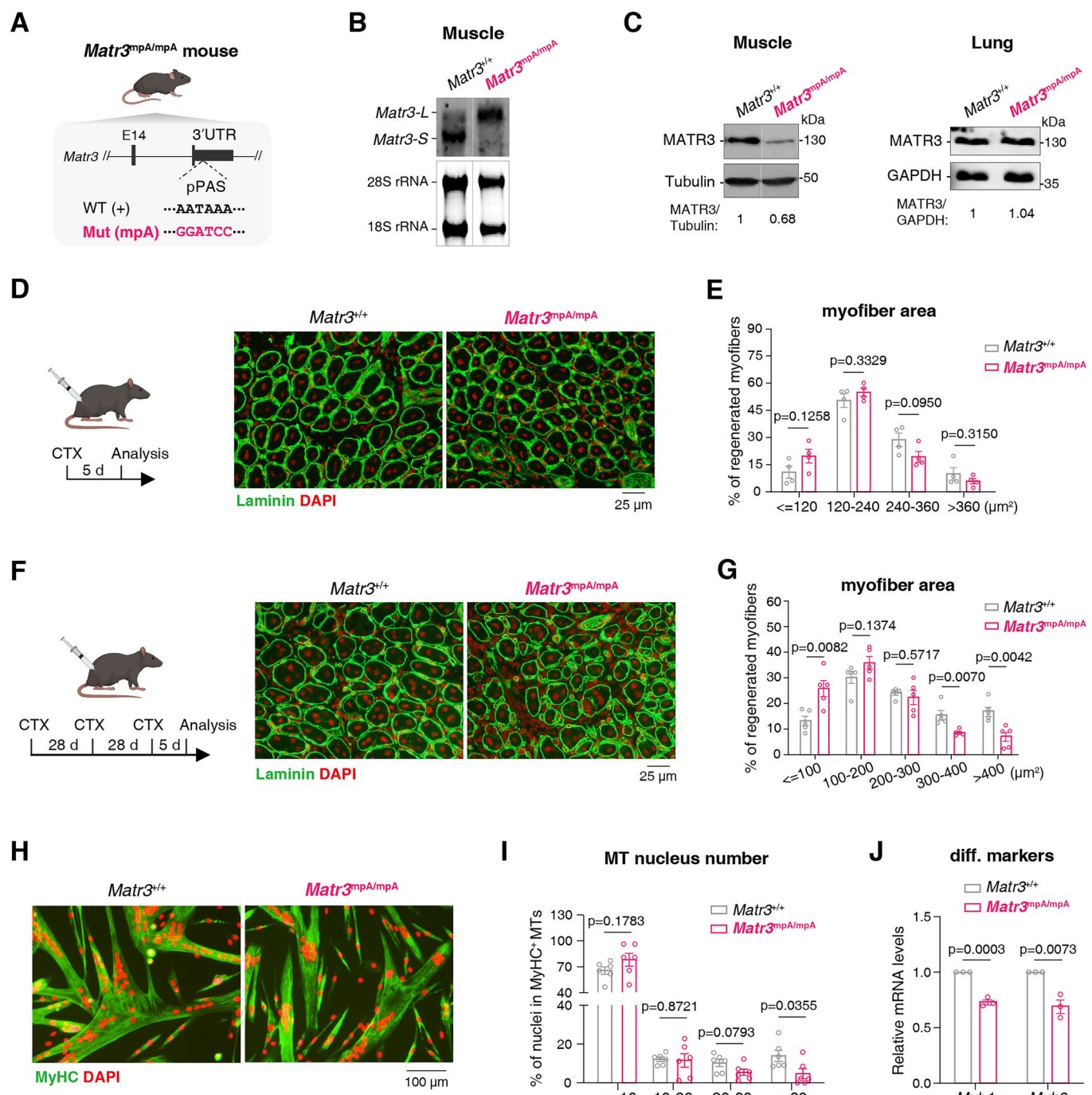

was performed using SYBR qPCR SuperMix (Novoprotein, E096-01B) according to the manufacturer's protocol. Primer sequences are provided in Appendix Table S4.

## Extraction of nuclear and cytoplasmic RNA

SCs and MTs were washed with cold 1× PBS and detached from culture plates using 0.25% Trypsin-EDTA (1×) (Gibco, 25200-072). The cells were collected into new tubes, and total RNA was extracted using TRIzol. For nuclear and cytoplasmic RNA isolation, cells were washed with cold 1× PBS and resuspended in lysis buffer 1 (10 mM Tris-HCl at pH 8.0, 140 mM NaCl, 1.5 mM MgCl₂, 0.5% Igepal), followed by incubation on ice for 5 min. The suspension was then centrifuged at $300 \times g$ for 5 min. The supernatant (cytoplasmic fraction) was transferred to a new tube, and RNA was extracted using TRIzol. Remaining pellet was resuspended in lysis buffer 2 (10 mM Tris-HCl at pH 8.0, 140 mM NaCl, 1.5 mM MgCl₂, 0.5% Igepal, 0.5% sodium deoxycholate) and centrifuged again. The supernatant was discarded. This lysis step with buffer 2 was repeated once, and after removing the supernatant, nuclear RNA was extracted from the final pellet using TRIzol.

◄

**Figure 6.  Genetically mutating *Matr3* proximal pA site in mice impairs muscle regeneration.**

(A) Schematic design of *Matr3* proximal pA site mutant (*Matr3*^mpA/mpA) mouse model. (B) Northern blot to detect the levels of *Matr3* short (*Matr3-S*) and long (*Matr3-L*) 3′ UTR isoforms in skeletal muscle from *Matr3*^+/+ and *Matr3*^mpA/mpA mice. 28S and 18S rRNAs were shown as loading controls. The white/gray line indicates the boundary where irrelevant lanes from the same blot were cropped. (C) Western blot to examine *Matr3* protein level in skeletal muscle and lung. Normalized MATR3 levels in *Matr3*^+/+ and *Matr3*^mpA/mpA mice are indicated below. The white line indicates the boundary where irrelevant lanes from the same blot were cropped. (D) Scheme of the CTX injury strategy (left). Representative immunofluorescence staining images of Laminin (green) and DAPI (red) in TA muscle sections at 5 days after single CTX injury in *Matr3*^+/+ and *Matr3*^mpA/mpA mice (right). Scale bar, 25 μm. (E) Quantification of myofiber regeneration capacity in *Matr3*^+/+ and *Matr3*^mpA/mpA mice following single CTX injury (*n* = 4 paired littermates). Myofibers were categorized by fiber area. Error bars, mean ± SEM. *P*-values, two-sided unpaired student's t test. (F) Scheme of the multiple CTX injuries strategy (left). Representative immunofluorescence staining images of Laminin (green) and DAPI (red) in TA muscle sections at 5 days after third CTX injury in *Matr3*^+/+ and *Matr3*^mpA/mpA mice (right). Scale bar, 25 μm. (G) Quantification of myofiber regeneration capacity in *Matr3*^+/+ and *Matr3*^mpA/mpA mice following multiple CTX injuries (*n* = 5 paired littermates). Myofibers were categorized by fiber area. Error bars, mean ± SEM. *P*-values, two-sided unpaired student's t test. (H) Representative immunofluorescence staining images of MyHC (green) and DAPI (red) in MTs differentiated from *Matr3*^+/+ SCs and *Matr3*^mpA/mpA SCs. Scale bar, 100 μm. (I) Quantification of myotube formation efficiency of SCs isolated from *Matr3*^+/+ and *Matr3*^mpA/mpA littermates. MyHC^+ myotubes were categorized by nuclear numbers. Data were obtained from six images across three paired littermates. Error bars, mean ± SEM. *P*-values, two-sided unpaired student's t test. (J) RT-qPCR to measure the mRNA levels of *Myh1* and *Myh3* in MTs differentiated from *Matr3*^+/+ SCs and *Matr3*^mpA/mpA SCs (*n* = 3). Bars represent RNA abundance normalized to *Gapdh*. Error bars, mean ± SEM. *P*-values, two-sided unpaired student's t test. Source data are available online for this figure.

## Quantseq

QuantSeq was performed using Lexogen's QuantSeq 3′ mRNA-Seq V2 Library Prep Kit FWD according to the manufacturer's protocol (Moll et al, 2014). Briefly, 500 ng of total RNA from SC or MT samples were reverse transcribed using an oligo-dT primer to initiate first-strand cDNA synthesis. After removal of the RNA template, second-strand synthesis was carried out using a random primer to generate double-stranded cDNA. The double-stranded library was then purified and amplified. Final libraries (200–500 bp) were used for deep sequencing.

## RNA-seq

For RNA-sequencing, 5 μg of nuclear and cytoplasmic RNAs from SCs or MTs were used for poly(A) + RNA selection. TruSeq Stranded Total RNA Sample Prep Kit (Illumina) was performed to generate stranded cDNA libraries according to the manufacturer's protocol. The cDNA libraries followed by sequencing on an Illumina Hiseq 2000 using a single-read protocol of 100 cycles with v3 chemistry.

## Polysome profiling

Polysome profiling was performed as previously described (Kang et al, 2023b). Briefly, C2C12 myoblasts and myotubes were homogenized in extraction buffer (100 mM NaCl, 50 mM Tris-HCl at pH 7.5, 5 mM MgCl$_2$, 1% Triton X-100, and 100 μg/ml cycloheximide). The homogenates were centrifuged at $13,000 \times g$ for 2 min to remove insoluble material. The resulting supernatants were carefully layered onto 10–60% sucrose gradients and centrifuged in an SW41 rotor at 38,000 rpm for 2 h at 4 °C. Twelve fractions were collected using a Piston Gradient Fractionator (Biocomp, Fredericton, Canada). RNA from fractions 7 through 12, corresponding to polysome-associated RNAs, was extracted and used for library preparation and deep sequencing.

## Northern blot analysis

Northern blot analysis was performed using Dig Northern Starter Kit (Roche, 12039672910) according to the manufacturer's protocol. In brief, 10 μg of total RNAs were separated by 1%

formaldehyde agarose gel, and transferred to a positively charged nylon membrane. Membrane then was hybridized with a probe to exons 12–15 of *Matr3*. Following washing steps to remove excessive probes, the bound probe was detected using chemiluminescent substrates.

## Western blot analysis

For cell protein sample preparation, cells were collected and lysed with 1× protein loading buffer (125 mmol/L Tris-HCl at pH 6.8, 4% SDS, 20% glycerol, 0.004% bromophenol blue, 20 mmol/L DTT) followed by denaturation at 95 °C for 10 min; for mouse tissue protein sample preparation, tissues were homogenized in ice-cold 1× RIPA buffer supplemented with PMSF, then the appropriate volumes of post-homogenization supernatant were mixed with 5× protein loading buffer and denatured at 95 °C for 10 min. Prepared protein samples were subsequently separated by bis-Tris SDS-PAGE and transferred onto polyvinylidene fluoride membranes, which were then blocked with 5% non-fat milk for 1 h at room temperature (RT). The membranes were incubated overnight at 4 °C with appropriately diluted primary antibodies, followed by a 1 h incubation at RT with secondary antibodies, and protein signals were subsequently visualized using Pierce ECL Substrate.

## Luciferase assays

HEK293 cells were co-transfected with Luc-WT or Luc-Mut luciferase reporter plasmids and miRNA expression plasmids, then cultured for 48 h. Luciferase activities were quantified with Dual-Luciferase Reporter Assay System (Promega, E1910) according to the manufacturer's protocol. Luc-WT and Luc-Mut plasmids were transfected separately in C2C12 myoblasts, followed by measurement of luciferase activities 24 h after transfection, or differentiation induction and measurement of luciferase activities at day 2.

## Immunofluorescence analysis

All procedures were performed at room temperature (RT). Differentiated myotubes were fixed with 4% paraformaldehyde for 15 min, washed with 1× PBS, and then permeabilized with 0.1% Triton X-100 for 15 min followed by another 1× PBS wash. The myotubes were then incubated with the primary antibody against Myosin Heavy Chain

## A

**% of diff-induced 3′UTR shortening**

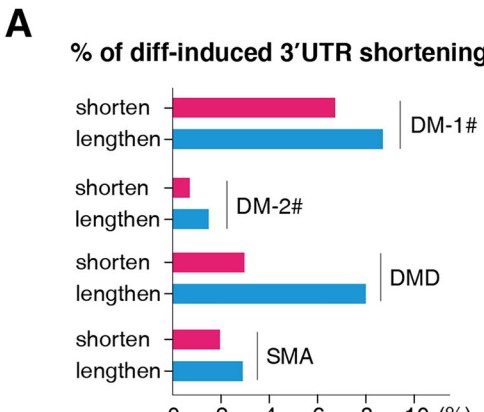

## B

**BP of DMD_APA & diff_shortening**

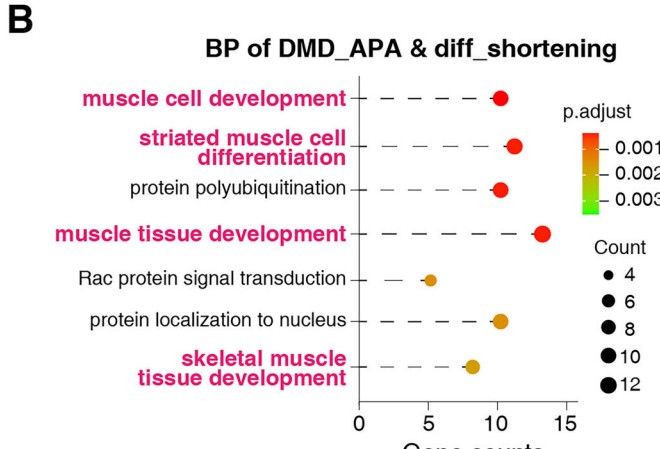

## C

**diseases_lengthening & diff_shortening**

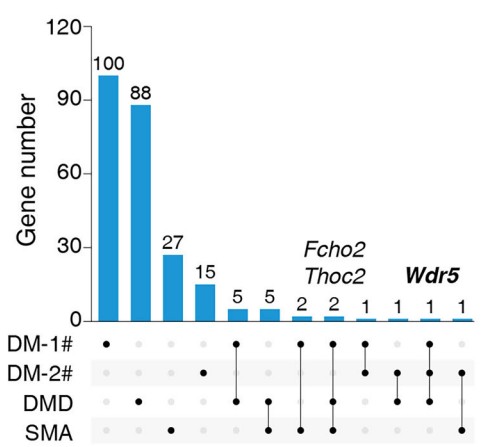

## D

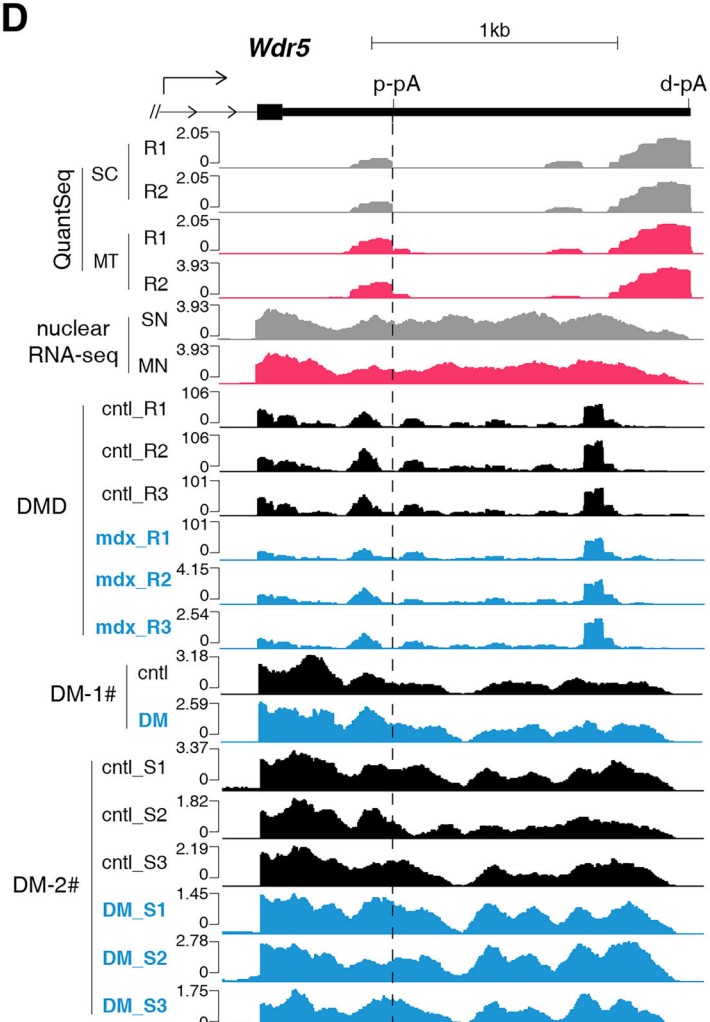

**Figure 7. Dysregulated 3′UTR shortening in muscle/neuromuscular diseases.**

(A) Bar plot showing the percentage of differentiation-induced 3′UTR-shortened RNAs that exhibiting significant APA changes (shorten: magenta; lengthen: blue) in different diseases, based on public data ((Meijboom et al, 2021; Batra et al, 2017; Georgieva et al, 2022) and GSE85984)). The labels DM-1# and DM-2# denote two independent datasets. (B) Dot plot showing Gene Ontology (Biological Process, BP) terms enriched in genes exhibiting 3′UTR shortening upon differentiation that also display APA changes in DMD mice. P-values were calculated using a hypergeometric test and adjusted for multiple hypothesis testing via the Benjamini-Hochberg method. (C) Upset plot showing the intersection of the indicated genes among four muscle disease groups. (D) Screenshot of QuantSeq, nuclear RNA-seq, and disease-associated RNA-seq signals of *Wdr5* 3′UTR.

(MyHC; 1:500) in blocking buffer (1× PBS, 0.1% Triton X-100, 2 mg/mL BSA) for 1 h. After washing with 1× PBS, the myotubes were incubated with Alexa Fluor 488- or 546-conjugated anti-mouse secondary antibodies (1:1000) in the same blocking buffer for 1 h. Nuclei were counterstained with DAPI. Images were captured using a DP72 charge-coupled device camera on an IX71 microscope, operated with DP-BSW software from Olympus. ImageJ and Acrobat software were bused to analyze the collected images.

For immunofluorescence staining of TA muscles cross-sections, the procedure was identical to that used for differentiated myotubes, except that the primary antibody used was anti-Laminin. Images were acquired on Olympus BX51 fluorescence microscope, and analyzed with ImageJ software.

## Muscle injury and cryosectioning

Muscle injury was performed as described (Wang et al, 2020). Briefly, 50 μL of 10 μM cardiotoxin (CTX) (Sigma, 217503) was injected intramuscularly into the tibialis anterior (TA) muscle of isoflurane-anesthetized mice. For the multiple injury protocol, two additional CTX injections were administered at 28-day intervals following the previous injury.

On the day of analysis, mice were euthanized, and TA muscles were harvested and embedded in Optimal Cutting Temperature compound, then rapidly frozen in liquid nitrogen for 30 s. Embedded muscles were transversely sectioned at a thickness of 10 μm. Sections were either processed immediately for experimental analysis or stored at −80 °C for long-term preservation.

## EdU incorporation analysis

EdU incorporation analysis was performed using Cell-Light EdU Apollo 567 In Vitro Kit (RIBOBIO, C10310-1) according to the manufacturer's instructions. Briefly, EdU was added to the SC culture medium at a final concentration of 50 μM and incubated for 2 h. SCs were then washed twice with 1× PBS, fixed with 4% PFA at RT for 30 min, and permeabilized with 0.5% Triton X-100 in 1× PBS for 10 min. The cells were subsequently incubated with the Apollo reaction buffer for 30 min. After washing, nuclei were counterstained with DAPI. Cell counting was performed using ImageJ software.

## Grip strength tests

Forelimb grip strength in littermate mice was assessed using a grip strength meter. For each mouse, five sequential measurements were obtained. The highest and lowest values were excluded, and the mean of the remaining three measurements was calculated to determine grip strength. Final grip strength values were normalized to body weight.

## Rotarod tests

The day before the experiment, littermate mice are trained on an accelerating rotarod at 5 rpm for 3 min twice, followed by 10 rpm for 3 min twice. On the following day, mice are tested on an accelerating rotarod starting at 5 rpm and accelerating to 40 rpm over 300 s. Each mouse undergoes five trials per day, with a minimum rest period of 15 min between trials. The time to fall from the rod is recorded for each mouse.

## Treadmill tests

Treadmill test were performed as previously described with minor modification (Wang et al, 2020). Littermate mice were first trained two days of an acclimation phase consisting of 5 min of 0 m/min and 8 min at 10 m/min. The next day, mice were placed on a treadmill, the speed of which was increased from 0 to 10 m/min over a period of 5 min and then stay at 10 m/min for an additional 10 min. The treadmill speed then increased (1 m/min every 1 min) to a maximum speed of 25 m/min until exhaustion. Mice were considered to be exhausted when the animal's hindlimbs remained on the electric grid for more than 10 s.

## Data analysis

### QuantSeq analysis

QuantSeq FWD reads were trimmed using Cutadapt (v2.6) (Martin, 2011) to remove adapters, polyT sequences from the 5′ end of Read1, and polyA sequences from the 3′ end of Read2. The trimmed reads were mapped to a combined index of the mm10 genome and ERCC RNA spike-ins using STAR (v2.7.10) (Dobin et al, 2013). Strand-specific bigWig coverage tracks were generated using deepTool (Ramírez et al, 2016). For the forward strand, the following parameters were used: '--filterRNAstrand forward --binSize 1', and for the reverse strand: '- filterRNAstrand reverse --binSize 1'. MAAPER (Li et al, 2021) was employed to predict PASs and analyze differential APA changes. Genes with a minimum of 50 aligned reads were included in the APA analysis. APA change was considered lengthen if the RED (Relative Expression Difference) exceeded 1.2 and shorten if the RED was less than five-sixths (i.e., <0.83). Statistically significant APA events were identified using a false discovery rate (FDR)-adjusted $p$-value < 0.05 (Fisher's exact test).

For differential gene expression analysis, gene exon read counts were quantified using featureCounts from the Subread (Liao et al, 2013). And DESeq2 (Love et al, 2014) was used to normalize read counts based on ERCC RNA spike-ins and to identify significantly differentially expressed genes (adjusted $p$-value < 0.05 and $|\log_2(\text{FC})| \geq 0.6$).

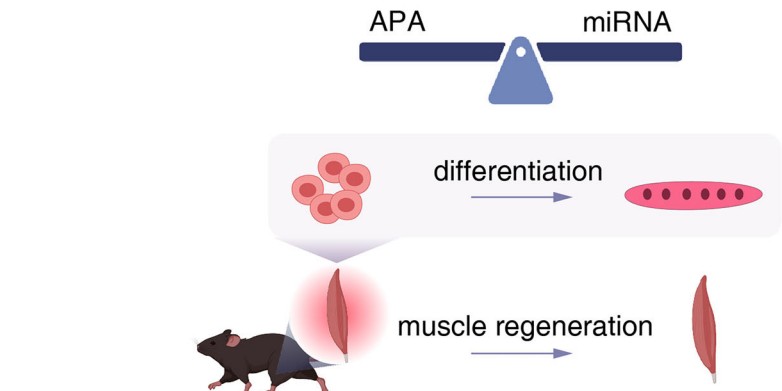

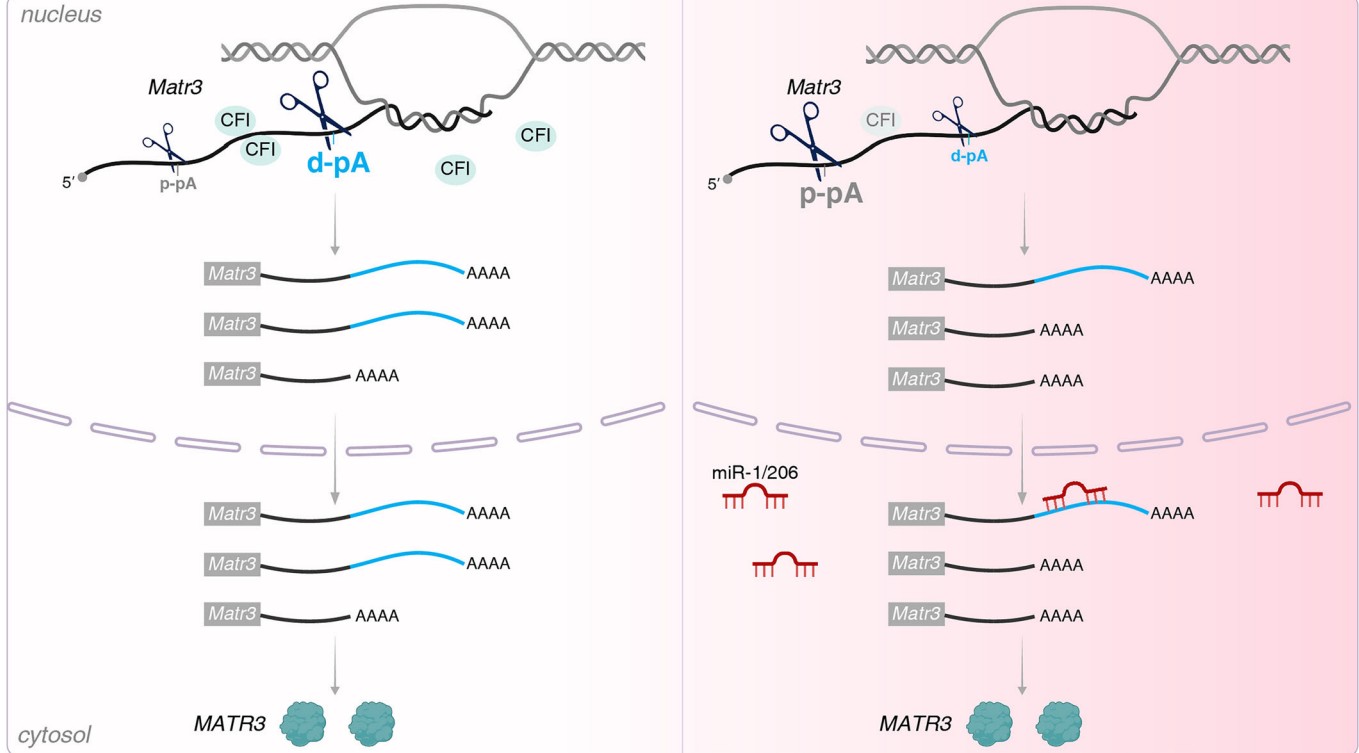

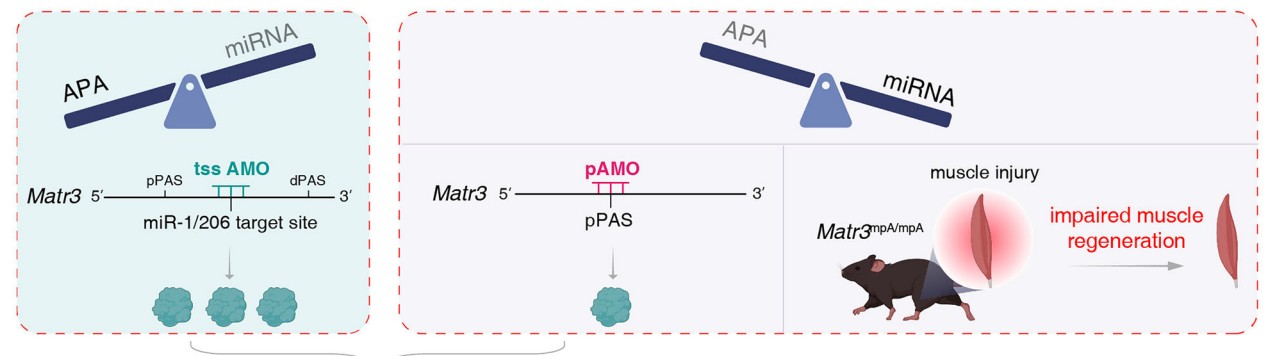

◀ **Figure 8. A model for APA and miRNA to counteractively regulate *Matr3* expression.**

During skeletal muscle differentiation, reduced CFI expression shifts polyadenylation toward *Matr3* proximal pA site, thereby facilitating *Matr3* evasion of miR-1/206-mediated repression. These two mechanisms counteractively regulate *Matr3* expression, which is essential for muscle differentiation and regeneration. Detailed description is included in the discussion.

## Nuclear and cytoplasmic RNA-seq analysis

For RNA-seq data, single-end reads were processed with Cutadapt (v1.18) (Martin, 2011) to trim adapters and low-quality sequences, retaining reads longer than 50 nt for subsequent mapping to rRNA sequences with STAR (v2.7.10) (Dobin et al, 2013). Reads not aligning to rRNA were further mapped to the mm10 genome using STAR (v2.7.10) (Dobin et al, 2013). APAlyzer (Wang and Tian, 2020) was used to identify APA events. Only APA isoforms with read count greater than 5 in at least one pair of comparing samples were used. The two most abundant APA isoforms based on PolyA_DB database (Wang et al, 2018) (https://exon.apps.wistar.org/PolyA_DB/v3/) for each gene were analyzed. Statistically significant APA events were identified using a false discovery rate (FDR)-adjusted $p$-value < 0.05 (Fisher's exact test).

For RNA export analysis, the cytoplasm-to-nucleus (C/N) ratio per gene was calculated as the ratio of cytoplasmic RPKM values to nuclear RPKM values.

Differentially expressed genes in SCs and MTs were determined by GFOLD (Feng et al, 2012), and heatmaps of 3′ processing factors were ranked based on GFOLD values.

## Polysome profiling analysis

Processing pipeline for polysome profiling data was consistent with that of RNA-seq analysis. Genes were quantified in both polysome fraction and input samples as reads per kilobase million (RPKM) using featureCounts from the Subread (Liao et al, 2013). Polysome enrichment (P.E.) was defined as the ratio of polysome fraction RPKM to input RPKM.

## 3′ READS data analysis

Publicly available data of 3′ READS for KD 3′ processing factors was downloaded from GSE62001 (Li et al, 2015). Processing of 3′ READS data was carried out as previously described (Hoque et al, 2013; Chen et al, 2019). Briefly, Reads were mapped to the mm10 using STAR (v2.7.10) (Dobin et al, 2013). Reads with ≥2 unaligned Ts at the 5′ end were used to identify pAs. Two most abundant pA isoforms in the 3′UTR of the last exon were selected for analysis. Significant APA events were those with $p$-value < 0.05 (Fisher's exact test).

## UGUA frenquency analysis

The proximal and distal pA sites within each gene, identified through QuantSeq, were utilized to define alternative untranslated regions (aUTRs). UGUA motifs were screened within these aUTRs. The frequency of UGUA motifs was calculated by normalizing the count of UGUA motifs to the length of aUTRs.

## CFI-68 tRIP-seq binding analysis

Publicly available CFI-68 tRIP-seq data was downloaded from GSE161602 (Kawachi et al, 2021). The analysis was performed as previously described (Masuda et al, 2020). Briefly, raw sequencing reads were processed to remove adapters and low-quality sequences using Cutadapt (v1.18) (Martin, 2011), discarding reads shorter than 18 nucleotides. Mapping was first performed against rRNA sequences with STAR (v2.7.10) (Dobin et al, 2013). Reads not aligning to rRNA were further mapped to the mm10 genome using STAR with default parameters. Multiply mapped reads were filtered out. Duplicates of reads uniquely mapped to the mouse genome were removed by Picard (v2.27.4). Finally, to identify significant peaks of CFI-68 binding, we used MACS (v2.2.7.1) (Zhang et al, 2008) with the following parameters '–f BAM––nomodel––shiftsize 25'.

## AGO2 CLIP-seq peak analysis

Publicly available AGO2 CLIP-seq data was downloaded from GSE57596 (Zhang et al, 2014). The 5′ barcode (NNNT) and 3′ adapter sequence (CTCGTATGCCGTCT) were removed from reads by Cutadapt (v1.18) (Martin, 2011), and the trimmed reads longer than 20 nt were mapped to the mm10 using Bowtie2 (Langmead and Salzberg, 2012) with the following parameters '--very-sensitive -rdg 5,2 --score-min L, -0.6, -0.7'. The peaks were called using the MACS (v2.2.7.1) (Zhang et al, 2008). AGO2 CLIP-seq peaks were further screened within aUTRs, and peaks exhibiting a normalized coverage FC > 2 between MTs and MBs were classified as increased AGO2 binding peaks.

## miRNA targeting analysis

The candidate myomiR target sequences are presented in Appendix Table S5. These sequences were identified and aligned in the regions of AGO2 binding peaks located in aUTRs using custom Python scripts.

## Conservation analysis

The conservation levels of myomiR target sequences, pPASs, and dPASs, along with their flanking regions, were evaluated using the 60-way vertebrate conservation Phastcons (Siepel et al, 2005) bigwig track from the UCSC Genome Browser database (mm10.60way.phastCons.bw).

## APA analysis in muscle diseases

Publicly available muscle disease-associated RNA-seq data from GSE168984 (Georgieva et al, 2022), GSE100943 (Batra et al, 2017), GSE150563 (Meijboom et al, 2021) and GSE85984 were downloaded. APA analyses of these data were consistent with those from the nuclear and cytoplasmic RNA-seq analysis. Statistically

significant APA events were identified using a false discovery rate (FDR)-adjusted $p$-value < 0.05 (Fisher's exact test).

## Functional enrichment analysis

Gene Ontology (GO) and Kyoto Encyclopedia of Genes and Genomes (KEGG) enrichment analyses were carried out by clusterProfiler package of R (Yu et al, 2012).

## Quantification and statistical analysis

Quantification of Microscopy images were performed using ImageJ and Acrobat software. Quantification of western blots were performed using ImageJ. Statistical significance was calculated using either GraphPad Prism or RStudio. Details of the statistical methods are described in the figure legends.

## Data availability

All sequencing data generated in this study have been deposited in the Gene Expression Omnibus (GEO) with the accession numbers GSE287180 (QuantSeq), GSE287801 (nuclear and cytoplasmic RNA-seq), and European Nucleotide Archive (ENA) with the accession number PRJEB46352 (Polysome-profiling seq) (https://www.ebi.ac.uk/ena/browser/view/PRJEB101157).

The source data of this paper are collected in the following database record: biostudies:S-SCDT-10_1038-S44318-025-00663-2.

## Peer review information

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

## Acknowledgements

We are grateful to Dr. Lili Han and Dr. Minzhi Jia for her technical advice for SC isolation and muscle regeneration. We thank Dr. Bin Tian and Dr. Chuwei Zhong for advices on the analysis of *Matr3* pAs. We thank Dr. Qiang Zhang for technical advices on northern blot. We appreciate Dr. Jing Fan and Cheng lab members for critical comments and valuable discussion. We thank the cell biology core facility and the animal core facility at CEMCS, SIBCB. The work was supported by the National Key R&D Program of China (2022YFA1303300), the Strategic Priority Research Program of the Chinese Academy of Science (XDB0570100), National Natural Science Foundation of China (32230022, 32430023) and the Open Research Fund of Beijing Advanced Center of RNA Biology (BEACON) at Peking University.

## Author contributions

**Yi Zhu**: Data curation; Investigation; Writing—original draft. **Jianshu Wang**: Data curation; Investigation. **Deng Tong**: Software; Formal analysis. **Peixuan Jia**: Software; Formal analysis. **Suli Chen**: Validation; Investigation. **Yangyang Li**: Software. **Jiaying Fu**: Investigation. **Qiming Li**: Formal analysis. **Ping Hu**: Resources; Supervision. **Yu Zhou**: Supervision. **Hong Cheng**: Conceptualization; Supervision; Funding acquisition; Visualization; Writing—original draft; Project administration; Writing—review and editing.

Source data underlying figure panels in this paper may have individual authorship assigned. Where available, figure panel/source data authorship is listed in the following database record: biostudies:S-SCDT-10_1038-S44318-025-00663-2.

## Disclosure and competing interests statement

The authors declare no competing interests.

# Expanded View Figures

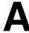

## A

**KEGG of 3′UTR lengthening**

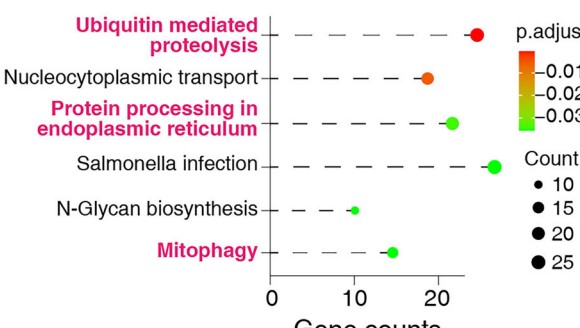

**KEGG of 3′UTR unchange**

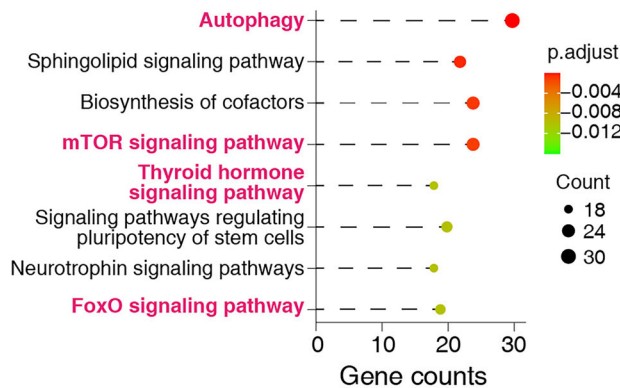

## B

**APA change (cytoplasmic polyA RNA-Seq)**

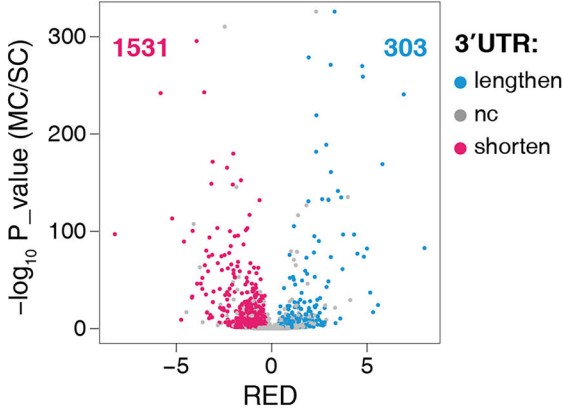

**Figure EV1.  Preferential 3′UTR shortening during satellite cell (SC) differentiation.**

(A) KEGG pathway enrichment analysis of genes with lengthened (left) and unchanged (right) RNA 3′UTRs. *P*-values were calculated using a hypergeometric test and adjusted for multiple hypothesis testing via the Benjamini-Hochberg method. (B) Volcano plot showing the APA changes in MC versus SC. APA changes were quantified using RED (relative expression difference, $\Delta\log_2[\text{d-pA/p-pA}]$) (Wang and Tian, 2020). Statistically significant APA events were identified using a false discovery rate (FDR)-adjusted *p*-value < 0.05 (Fisher's exact test). Genes exhibiting significant 3′UTR shortening and lengthening are indicated in magenta and blue. Gene counts are shown.

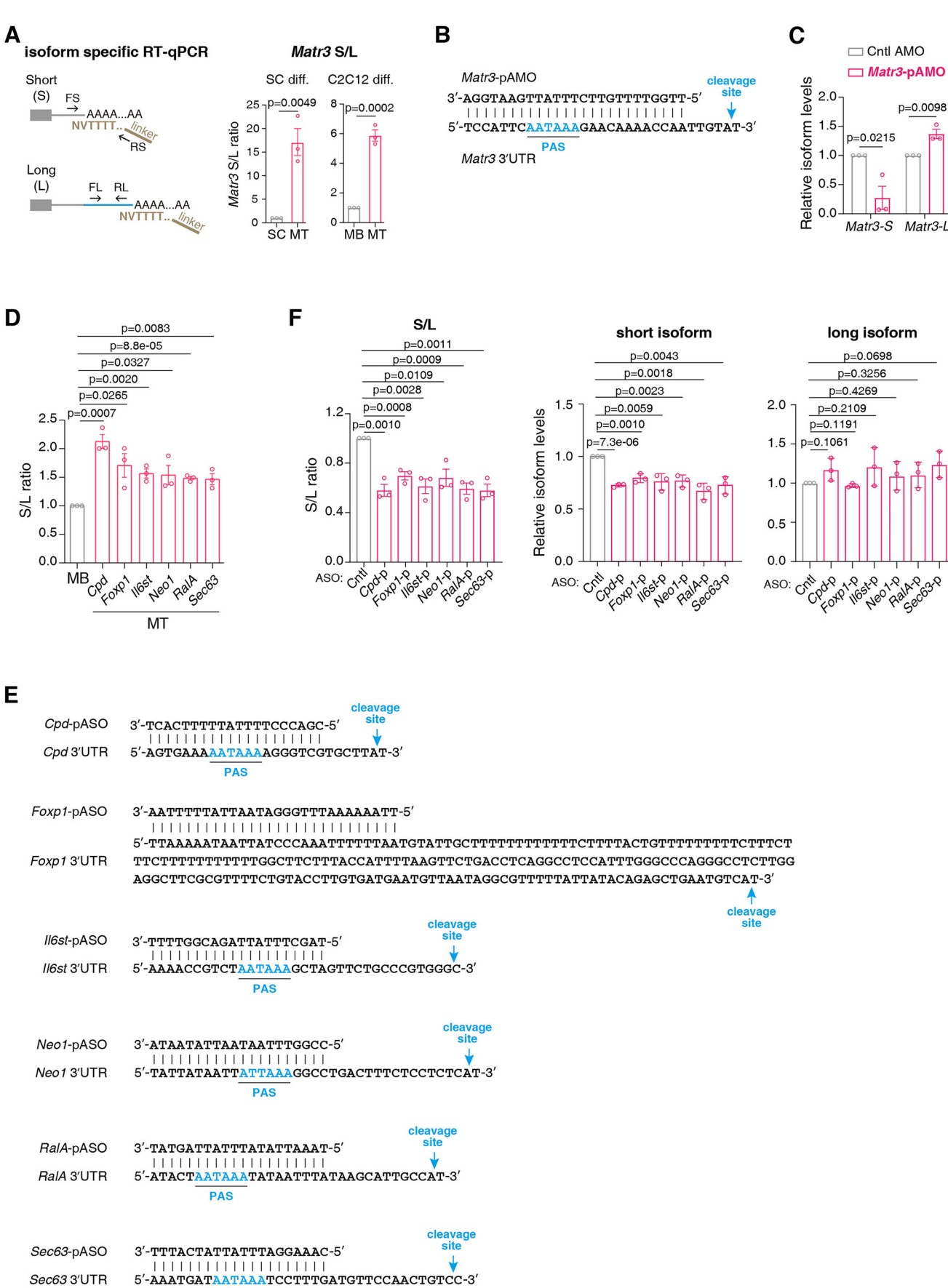

**Figure EV2.   Inhibition of 3′UTR shortening impairs myogenic differentiation.**

(A) Schematic of the isoform-specific RT-qPCR strategy (left). Reverse transcription primers are indicated in bronze. FS, forward primer of short 3′UTR isoform; RS, reverse primer of short 3′UTR isoform; FL, forward primer of long 3′UTR isoform; RL, reverse primer of long 3′UTR isoform. RT-qPCR to measure the ratio of short to long (S/L) *Matr3* isoforms during SC differentiation and MB differentiation (right) ($n = 3$). Error bars, mean ± SEM. *P*-values, two-sided unpaired student's t test. (B) Sequences of the *Matr3*-pAMO and its complementary target within the 3′UTR of *Matr3* are shown. The polyadenylation signal (PAS) and cleavage sites are indicated. (C) Isoform-specific RT-qPCR to measure short and long isoform levels of *Matr3* in MTs differentiated from SCs treated with either control AMO (Cntl AMO) or *Matr3*-pAMO ($n = 3$). Bars represent RNA abundance normalized to *Gapdh*. Error bars, mean ± SEM. *P*-values, two-sided unpaired student's t test. (D) Isoform-specific RT-qPCR to measure the short to long (S/L) isoform ratio of indicated genes in MBs and differentiated MTs ($n = 3$). Error bars, mean ± SEM. *P*-values, two-sided unpaired student's t test. (E) Sequences of the indicated ASOs and their complementary 3′UTR regions. PAS and cleavage sites are indicated. (F) Isoform-specific RT-qPCR to measure the S/L ratios, short and long isoform levels of the indicated genes in MTs differentiated from MBs treated with ASOs blocking the proximal pA site of the corresponding gene ($n = 3$). For results of short and long isoform levels, bar represent RNA abundance normalized to *Gapdh*. Error bars, mean ± SEM. *P*-values, two-sided unpaired student's t test. Source data are available online for this figure.

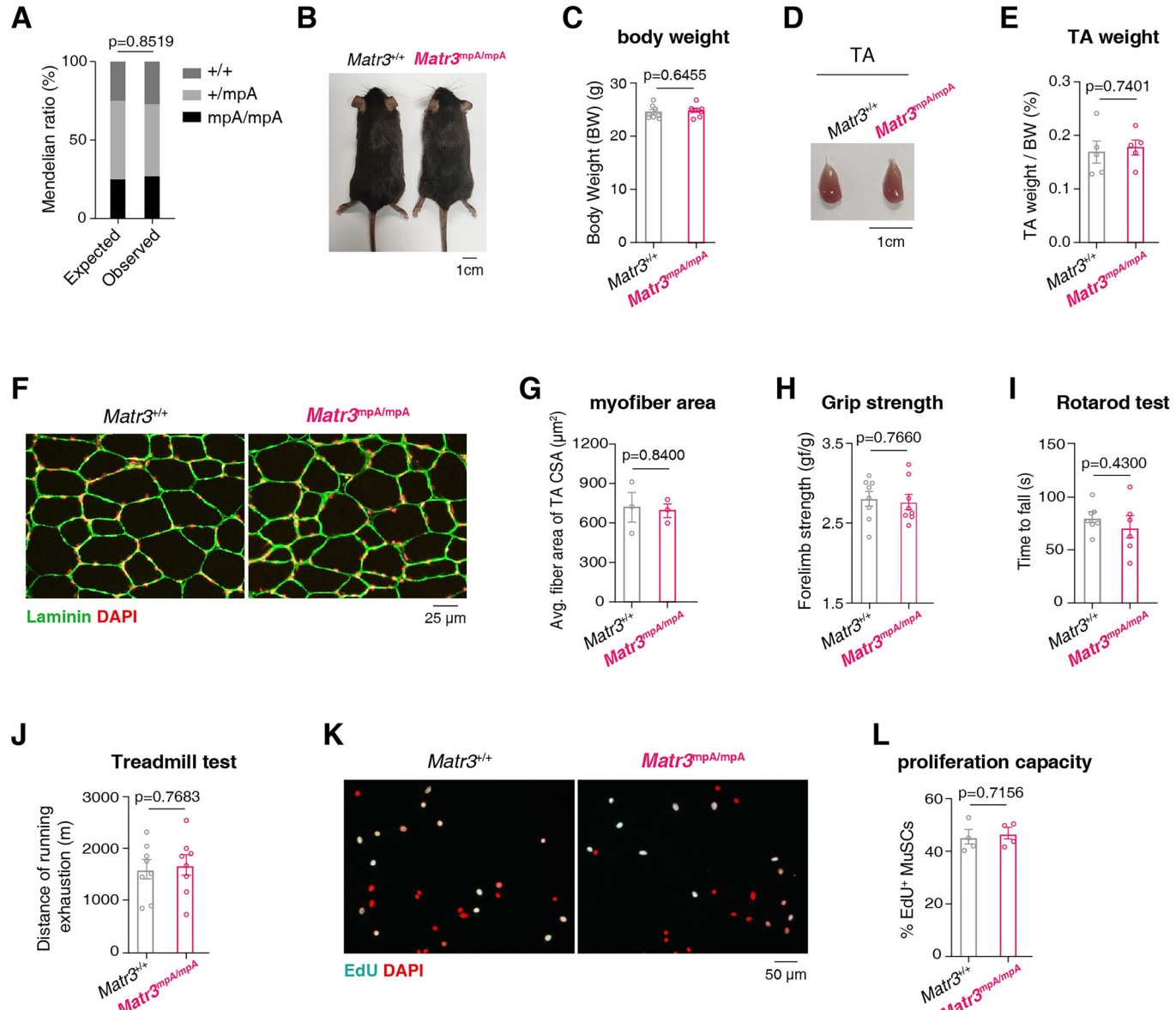

**Figure EV3.**   *Matr3*^mpA/mpA mice exhibited normal skeletal muscle development and motor function.

(A) *Matr3*^mpA/mpA mice were born at the expected Mendelian ratio ($n = 108$). No significant differences between the expected and observed genotypic numbers, as determined by chi-squared test. (B) Representative photographs of 8-week-old *Matr3*^+/+ and *Matr3*^mpA/mpA littermates. Scale bar, 1 cm. (C) Bar plot quantifying the body weights of 8-week-old *Matr3*^+/+ and *Matr3*^mpA/mpA littermates ($n = 8$). Error bars, mean ± SEM. *P*-values, two-sided unpaired student's t test. (D) Representative photographs of TA muscles from *Matr3*^+/+ and *Matr3*^mpA/mpA littermates at 12 weeks of age. Scale bar, 1 cm. (E) Bar plot showing the ratio of TA muscle weight to body weight (BW) in *Matr3*^+/+ and *Matr3*^mpA/mpA littermates at 12 weeks of age ($n = 5$). Error bars, mean ± SEM. *P*-values, two-sided unpaired student's t test. (F) Representative immunofluorescence staining images of Laminin (green) and DAPI (red) of TA muscle cross-sections from *Matr3*^+/+ and *Matr3*^mpA/mpA littermates at 12 weeks of age. Scale bar, 25 µm. (G) Quantification of average cross-section area (CSA) of TA myofibers from *Matr3*^+/+ and *Matr3*^mpA/mpA littermates at 12 weeks of age ($n = 3$). Error bars, mean ± SEM. *P*-values, two-sided unpaired student's t test. (H) Forelimb grip strength (normalized to body weight) was measured in *Matr3*^+/+ and *Matr3*^mpA/mpA littermates at 12 weeks of age ($n = 8$). Error bars, mean ± SEM. *P*-values, two-sided unpaired student's t test. (I) Rotarod tests were assessed in *Matr3*^+/+ ($n = 7$) and *Matr3*^mpA/mpA ($n = 6$) littermates at 12 weeks of age. The time it takes for the mouse to fall off the rod was recorded. Error bars, mean ± SEM. *P*-values, two-sided unpaired student's t test. (J) Distances of running exhaustion were measured in *Matr3*^+/+ and *Matr3*^mpA/mpA littermates at 4-5 months of age ($n = 8$). Error bars, mean ± SEM. *P*-values, two-sided unpaired student's t test. (K) Representative images of EdU (cyan) incorporation in SCs isolated from *Matr3*^+/+ and *Matr3*^mpA/mpA littermates. DAPI (red) was used for nuclear counterstaining. Scale bar, 50 µm. (L) Quantification of SC proliferation capacity from *Matr3*^+/+ and *Matr3*^mpA/mpA littermates ($n = 4$). The ratio of EdU-positive (Edu^+) SCs to the total number of cells (DAPI) was calculated. Error bars, mean ± SEM. *P*-values, two-sided unpaired student's t test. Source data are available online for this figure.

## A

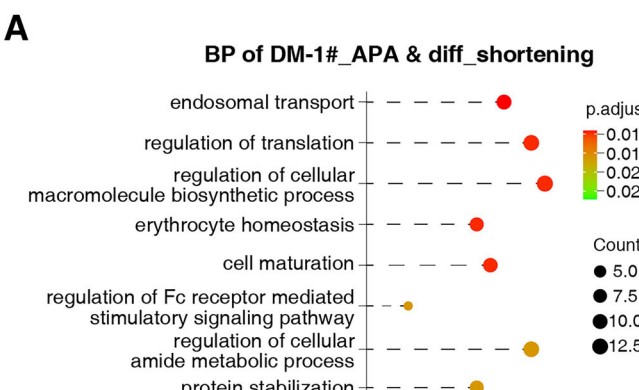

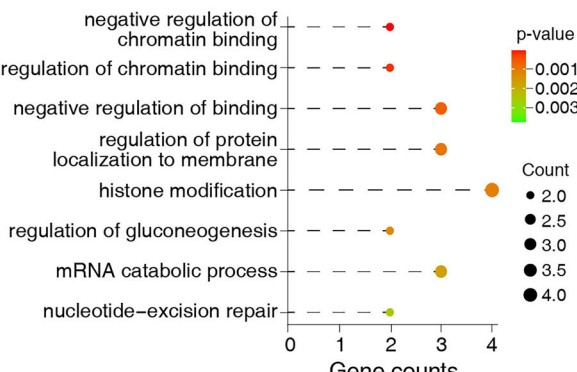

## B

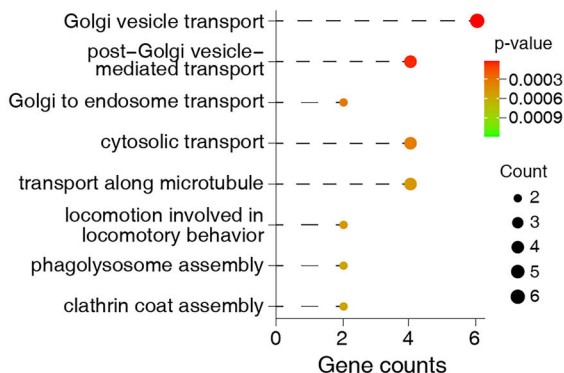

**Figure EV4.  Dysregulated 3′UTR shortening in muscle/neuromuscular diseases.**

(**A**) Dot plot showing Gene Ontology (Biological Process, BP) terms enriched in genes exhibiting 3′UTR shortening upon differentiation that also display APA changes in DM samples. (**B**) Dot plot showing Gene Ontology (Biological Process, BP) terms enriched in genes exhibiting 3′UTR shortening upon differentiation that also display APA changes in SMA samples. For dot plots in (**A**) and (**B**), *p*-values were calculated using a hypergeometric test and adjusted for multiple hypothesis testing via the Benjamini-Hochberg method.

