## [Peer Review File · The EMBO Journal]

3'UTR shortening alleviates miRNA repression of mRNAs critical for muscle stem cell differentiation

Yi Zhu, Jianshu Wang, Deng Tong, Peixuan Jia, Suli Chen, Yangyang Li, Jiaying Fu, Qiming Li, Ping Hu, Yu Zhou, and Hong Cheng

Corresponding author(s): Hong Cheng (hcheng@sibcb.ac.cn) , Yu Zhou (yu.zhou@whu.edu.cn), Ping Hu (hu_ping@gzlab.ac.cn)

Review Timeline:

Submission Date:	15th Jul 25
Editorial Decision:	2nd Sep 25
Revision Received:	30th Sep 25
Editorial Decision:	14th Nov 25
Revision Received:	18th Nov 25
Accepted:	27th Nov 25

Editor: Hartmut Vodermaier

Transaction Report:

Prof. Hong Cheng
Shanghai Institute of Biochemistry and Cell Biology, Center for Excellence in Molecular Cell Science, Chinese Academy of Sciences, Shanghai ; University of Chinese Academy of Sciences, China
Key Laboratory of RNA Innovation, Science and Engineering, Shanghai Key Laboratory of Molecular Andrology
320 Yue Yang Road
Shanghai 200031
China

2nd Sep 2025

Re: EMBOJ-2025-121889
3'UTR shortening alleviates miRNA repression for muscle stem cell differentiation

Dear Hong,

Thank you again for submitting your study on alternative polyadenylation roles in muscle stem cell differentiation to The EMBO Journal. We have now received comments from four referees with expertise in both of these areas, copied below for your information. As you will see, all referees are generally supportive of publication. Nevertheless, while referees 1-3 only have a certain number of specific points, referee 4 raises several more substantial queries that would need to be adequately responded to.

In light of these reports, we would be interested in considering this study further for publication in our journal. To clarify how the issues raised in the reports could be best addressed, I would however invite you to contact me with a tentative point-by-point response and revision plan already during the early stages of the revision work, so that we could discuss the feasibility and concrete requirements of particular revisions ahead of time. We would also be open to extending the revision deadline if that should be helpful. Our scooping protection policy means that competing manuscripts published while your work is under revision will not have a negative effect on our final decision.

Detailed information on preparing, formatting and uploading a revised manuscript can be found below and in our Guide to Authors. Thank you again for the opportunity to consider this work for The EMBO Journal, and I look forward to hearing from you in due time.

With kind regards,

Hartmut

9) To facilitate reproducibility and cross-laboratory adoption of methodologies, please structure the Materials & Methods section as outlined in our guide to authors, including a completed Reagents and Tools Table that can be downloaded from our author guidelines as well (<https://www.embopress.org/page/journal/14602075/authorguide#structuredmethods>).

10) Digital image enhancement is acceptable practice, as long as it accurately represents the original data and conforms to community standards. If a figure has been subjected to significant electronic manipulation, this must be clearly noted in the figure legend and/or the 'Materials and Methods' section. The editors reserve the right to request original versions of figures and the original images that were used to assemble the figure. Finally, we generally encourage uploading of numerical as well as gel/blot image source data; for details see: embopress.org/page/journal/14602075/authorguide#sourcedata

In the interest of ensuring the conceptual advance provided by the work, we recommend submitting a revision within 3 months (1st Dec 2025). Please discuss the revision progress ahead of this time with the editor if you require more time to complete the revisions. Use the link below to submit your revision:

Link Not Available

Referee #1:

In this manuscript by Zhu et al., the authors report 3'UTR shortening via alternative polyadenylation in muscle stem cell differentiation. The shortening trend coincides with increased expression of muscle-specific miRNAs (myomiRs). In line with these, they show 3'UTR shortening helps evasion of miRNA targeting for some genes. A critical case pursued in this work is *Matr3*, whose expression is upregulated and 3'UTR is shortened during differentiation. Suppression of proximal pA site usage of *Matr3* by antisense morpholino oligonucleotide (*Matr3*-pAMO) leads to increased ratio of long 3'UTR isoform abundance to that of short 3'UTR isoform and inhibits cell differentiation *in vitro*. Further, the authors show that loss-of-function mutation of the proximal pA site of mouse *Matr3* obviates its APA regulation and impaired *Matr3*-mediated muscle regeneration *in vivo*. The authors showed multiple lines of evidence supporting a causal role of CFI_m downregulation in 3'UTR regulation in myogenesis: (1) CFI KD induced 3'UTR shortening during differentiation; (2) both CFI-25 and CFI-68 expression are shown to drop at the mRNA and protein levels; (3) dPAS UGUA motif enrichment; (4) a luciferase reporter assay with the *Matr3* 3'UTR (WT vs. mutated CFI binding motifs) directly demonstrates that CFI binding to UGUA motifs is required to promote dPAS usage. Overall, the work presented in this paper was well carried out. The data support their conclusion that a subset of genes display 3'UTR shortening in muscle stem cell differentiation which impacts miRNA-mediated gene regulation. The findings reported in this paper are novel in the sense that previous myogenesis models *in vitro* were found to display a general 3'UTR lengthening. As such, the result of this work provides a fuller understanding of APA regulation in myogenesis, and perhaps cell differentiation in

general. I do, however, have a few comments and suggestions aimed to further strengthen this paper.

- Fig. 1D, all three groups of genes, including lengthen, nc and shorten appear to be downregulated, as indicated by the median values shown under the Violin plots. Does this mean genes with APA isoforms are generally downregulated or there is a data normalization issue?
- Fig. 2. The Matr3-pAMO seems to work well to inhibit p-pA usage. However, the authors show the S/L ratio only. Presumably, the pAMO could remain bound to the long isoform and lead to its stability or translation changes? The same issue goes to other genes shown in Fig. 2H. It would also be informative if the authors show the sequence to which the pAMO target and indicate where the cleavage site is.
- Figure 3C, because the data is frequency-based, Wilcoxon rank-sum test may not be appropriate. As such, while the p values satisfy the conventional threshold of 0.05, it's hard to discern if the three groups are truly significantly different.
- The finding that downregulation of CFI-25 and -68 is responsible for 3'UTR shortening is interesting. However, the causality would be better supported if re-expressing CFI-25/CFI-68 could reverse global 3'UTR shortening. In addition, a direct CFIm binding assay, such as eCLIP and RIP, would help strengthen CFIm occupancy at UGUA-decorated dPASs.
- It would be useful to provide some discussion on why CFI expression is downregulated.
- The greater binding of Ago2 to aUTRs of 3'UTR shortened genes is interesting. On the one hand it may indicate shortening of 3'UTR can evade miRNA targeting. On the other, it suggests that some of the shortening cases may be attributable to long isoform degradation. The authors need to discuss the possibility of latter and whether it may also contribute the APA isoform difference between different models, i.e., distinct miRNA targeting activities in different systems may impact the 3'UTR length profile differently.
- Further on the discussion of the difference between C2C12 model and primary muscle stem cells, the authors may want to comment on the differentiation stages, e.g., early vs. late. Pertinent to myogenesis, there is a cell fusion step during differentiation. Would it be possible that the 3'UTR length control is different before vs. after this step?

Referee #2:

In this manuscript, Zhu et al characterized mRNA alternative polyadenylation (APA) during myogenic differentiation, and elucidated the functional significance of key genes that displayed significant APA changes. Although there were previous studies that looked into this topic using cell line systems, this study used stem cells isolated from mice and was therefore more physiologically relevant and found new and interesting targets. Additionally they not only examined the molecular mechanisms for the observed APA changes, but also meticulously investigated the downstream functional impact using a variety of approaches, from antisense oligoes to mutant animal models. It is quite rare and refreshing to see such in-depth and comprehensive studies. I think this study is fully ready to be published in EMBO J as is.

Referee #3:

In the manuscript by Zhu, Wang, Tong, Jia et al. entitled "3'UTR shortening alleviates miRNA repression for muscle stem cell differentiation" the authors explore the interplay between 3'UTR length and targeting by muscle-enriched miRNAs (myomiRs) during muscle cell differentiation. They find results different than previous studies with cell lines and validate their results in vivo. This work provides new insights into the intricacies of muscle cell differentiation and repair.

The study includes a number of complementary approaches to validate the results obtained, including the generation and analysis of a mouse model to validate the findings in vivo with a focus on one key gene identified in their analysis, Matr3. These orthogonal approaches provide mechanistic insight.

As a rather minor point, the authors should carefully consider their language throughout as no genes are shortened or lengthened- the transcripts, or more precisely the 3'UTR of the transcripts, are shortened or lengthened. The use of the term shortened gene is short-hand to indicate that the gene encodes a transcript with an altered 3'UTR. The authors should just carefully consider the precision of the language used throughout. For example, in Figure 4I, the authors should really change the label as it currently reads "Shortened genes with increased Ago2 binding". The genes are not shortened and the Ago2 is actually not binding to genes.

The statistical tests employed to analyze the data are not described in sufficient details in either the Materials and Methods or the Figure Legends. The authors need to clearly indicate what type of statistical tests are performed for each analysis. Although p-values are provided throughout, the type of tests performed (appears primarily to be t-tests) are not indicated. In addition, results that are statistically significant are not indicated by asterisks as is typical. This makes the data more difficult for the reader to interpret. The authors should employ the conventional approach of indicating statistical significance and degree of statistical significance with *, **, ***.

Specific Comments:

The study starts with an analysis of the 3'UTR of transcripts performed in primary muscle satellite cells and in differentiated myotubes from mice. This analysis is different than prior studies of muscle cell differentiation, which have primarily relied on the mouse C2C12 cell line rather than primary cells. For the analysis shown in Figure 1C, the KEGG analysis of genes where the transcripts are shortened, the authors identify key categories critical for muscle differentiation. The text does not make clear whether the authors also performed a similar analysis with the set of genes that have lengthened 3'UTRs or those that do not change. As these are primary muscle cells, the terms identified are certainly likely to relate to muscle. The most robust analysis would focus on those terms enriched for the transcripts that show a shortened 3'UTR compared to other datasets- not just the shortened 3'UTR transcripts as an independent dataset. Perhaps the authors performed this analysis, but the point is not clear in the text or the figure.

Figure 2A, which indicates the reads for the Matr3 gene does not indicate the number of bases on the X-axis. This information should be included in the figure as shown markers on the gel image shown in Figure 2B. In Figure 2G and other such figures, labeling the Y-axis as 'Relative Transcript Abundance' would make the figure easier to interpret for the reader along with the main text of the manuscript without having to consult the figure legend, which does clearly indicate the analysis of transcript levels.

The data presented in Figure 2H-K does not seem to include the transcripts where the ASO treatment did NOT impair MT formation (beyond the proof of principle in Figure 2H that the ASO works to shift the ratio of short to long (S/L) 3'UTR in Figure 2H. Instead these controls are shown in Figure S2, but they should be incorporated into the primary data shown in the main text figures. The controls are critical to interpret the data presented.

Minor points:

For the broad readership of EMBO J., the authors should be sure to define all abbreviations including in the figures. Figure 1A includes the abbreviations SC and MT without definition in the figure (where there really is space) or, at a minimum, in the Figure Legend.

Really minor point:

The formatting of one of the sets of references on p. 8 has an error because the title of the one of the papers is included in the list of references.

Referee #4:

In this study, the investigators provided some interesting findings to show the 3' UTR shorting is associated with gene function and muscle differentiation.

When performing polyA RNA-seq, why they separated nucleus and cytoplasm portion? The difference in 3'UTR shorting is quite different in N. and C. compartments. Why? Why were these results so different from that of the QuantSeq analysis? (Fig. 1).

The authors cited Quintero-Rivera (Human Molecular Genetics 24: 2375-2389), and stated that "its short 3'UTR isoform is specifically detected in skeletal and cardiac muscle tissues", and "truncated Matr3 3'UTR exhibited abnormal motor functions". These statements are not true. The prior study described that "Matr3(Gt-ex13) heterozygotes exhibit incompletely penetrant BAV, CoA and PDA phenotypes similar to those in the human proband, as well as ventricular septal defect (VSD) and double-outlet right ventricle (DORV)". Those are not muscle expressing tissues and there is no defect in motor function.

They verified the 3'UTR shorting of Matr3 gene in C2C12 myoblast differentiation. They should do that for their other tested genes (Cpd, Foxp1, Il6st, Neo1, RalA, and Sec63) as well.

Fig. 3I, they used RT-PCR to document Luc-S/Luc-L ratio. Can they also provide quantitative measurement of the luciferase activities?

They found that "genes with lengthened or unchanged 3'UTRs, but not that of shorted, showed significant decreases in polysome association in MTs versus MBs (Figure 4)". It will be nice if they can show protein levels of representative targets.

Wasn't the AGO2 CLIP data from Zhang 2014 showed Ago2 binding to 3'UTRs of mitochondrial mRNAs?

The muscle phenotype data from the Matr3mpA/mpA (without injury) is not consistent with that of in vitro study, in which they showed that Matr3-pAMO affected SC differentiation. How do you reconcile this? Is matr3 (or different isoforms of the transcript) expressed in SCs, undifferentiated MBs and differentiated MTs with specificity?

They have used antisense morpholino oligonucleotide to block proximal pA site of the Matr3 transcripts. As we all know AMOs

could be non-specific, or off-target, they should utilize an independent assay to verify their findings.

The data presented in Fig 7 is very confusion. Muscle defect in DMD patients was caused by mutations of the dystrophin gene. How do those dystrophin mutations then lead to dysregulation of 3'UTR shorting?

The hypothetic models presented in Figs 7 and 8 are quite confusion, and overinterpreted their findings. The roles of myomiRs in muscle differentiation was never tested directly in the context of Matr3 3'UTR. They only presented that in the luciferase reporter assays performed in non-muscle 293T cells.

Minors:

Incorrect formatting or incomplete in reference citations:

(Tian & Manley, 2017; Lackford et al, 2014; Li et al, 2015; The Polyadenylation Factor CstF-64 Regulates Alternative Processing of IgM Heavy Chain Pre-mRNA during B Cell Differentiation, 1996; Martin et al, 2012)

The Polyadenylation Factor CstF-64 Regulates Alternative Processing of IgM Heavy Chain Pre-mRNA during B Cell Differentiation (1996) Cell 87: 941-952

Referee #1:

In this manuscript by Zhu et al., the authors report 3'UTR shortening via alternative polyadenylation in muscle stem cell differentiation. The shortening trend coincides with increased expression of muscle-specific miRNAs (myomiRs). In line with these, they show 3'UTR shortening helps evasion of miRNA targeting for some genes. A critical case pursued in this work is Matr3, whose expression is upregulated and 3'UTR is shortened during differentiation. Suppression of proximal pA site usage of Matr3 by antisense morpholino oligonucleotide (Matr3-pAMO) leads to increased ratio of long 3'UTR isoform abundance to that of short 3'UTR isoform and inhibits cell differentiation in vitro. Further, the authors show that loss-of-function mutation of the proximal pA site of mouse Matr3 obviates its APA regulation and impaired Matr3-mediated muscle regeneration in vivo. The authors showed multiple lines of evidence supporting a causal role of CFI downregulation in 3'UTR regulation in myogenesis: (1) CFI KD induced 3'UTR shortening during differentiation; (2) both CFI-25 and CFI-68 expression are shown to drop at the mRNA and protein levels; (3) dPAS UGUA motif enrichment; (4) a luciferase reporter assay with the Matr3 3'UTR (WT vs. mutated CFI binding motifs) directly demonstrates that CFI binding to UGUA motifs is required to promote dPAS usage. Overall, the work presented in this paper was well carried out. The data support their conclusion that a subset of genes display 3'UTR shortening in muscle stem cell differentiation which impacts miRNA-mediated gene regulation. The findings reported in this paper are novel in the sense that previous myogenesis models in vitro were found to display a general 3'UTR lengthening. As such, the result of this work provides a fuller understanding of APA regulation in myogenesis, and perhaps cell differentiation in general. I do, however, have a few comments and suggestions aimed to further strengthen this paper.

We thank this reviewer for his/her appreciation for our findings on the mechanism and function of 3'UTR shortening in myogenesis.

1) Fig. 1D, all three groups of genes, including lengthen, nc and shorten appear to be downregulated, as indicated by the median values shown under the Violin plots. Does this mean genes with APA isoforms are generally downregulated or there is a data normalization issue?

We thank the reviewer for this critical point. As the reviewer reasoned, we found a normalization issue arising from the initial sequencing depth-based analysis method, which led to the seemingly downregulation across all groups in original Fig. 1D. We

re-analyzed the data using ERCC spike-in controls for normalization. This refined analysis revealed that only the nc group exhibits moderate downregulation, while the lengthen and shorten groups remain largely unaltered. Thus, no apparent trend in 3'UTR length changes in affecting mRNA levels during myogenesis. We replaced the original Fig. 1D with the updated ERCC-normalized version in the revised manuscript (**revised Fig. 1D**).

2) *Fig. 2. The Matr3-pAMO seems to work well to inhibit p-pA usage. However, the authors show the S/L ratio only. Presumably, the pAMO could remain bound to the long isoform and lead to its stability or translation changes? The same issue goes to other genes shown in Fig. 2H. It would also be informative if the authors show the sequence to which the pAMO target and indicate where the cleavage site is.*

Thank you for raising this important point. To test the possibility that the decreased S/L ratio might result from a decline in long isoform expression after AMO/ASO treatment, we have now examined the abundance of both the short and long isoform levels. These data clearly demonstrate AMO/ASO treatment resulted in decreased level of the short isoform, rather than a decline in the long isoform. We have included these data in the revised manuscript and figures (**Fig. EV2C** for *Matr3* and **Fig. EV2F** for other genes).

As suggested, we have provided the target sequence of AMO/ASO and indicated the cleavage site (**revised Fig. EV2B, E**).

3) *Figure 3C, because the data is frequency-based, Wilcoxon rank-sum test may not be appropriate. As such, while the p values satisfy the conventional threshold of 0.05, it's hard to discern if the three groups are truly significantly different.*

Thank you for this insightful comment. As suggested, we have re-analyzed the data using a t-test. We found that mRNAs with shortened 3'UTRs exhibited a significantly higher UGUA motif frequency compared to those with lengthened 3'UTRs. We further categorized the shortened mRNAs into two groups depending on whether CFI-68 KD led to their 3'UTR shortening or not. mRNAs that were sensitive to CFI-68 KD showed a higher UGUA motif frequency than those do not. We have replaced the original Fig. 3C with the updated one (**revised Fig. 3C**).

4) *The finding that downregulation of CFI-25 and -68 is responsible for 3'UTR*

shortening is interesting. However, the causality would be better supported if re-expressing CFI-25/CFI-68 could reverse global 3'UTR shortening. In addition, a direct CFI binding assay, such as eCLIP and RIP, would help strengthen CFI occupancy at UGUA-decorated dPASs.

We appreciate the interest of the reviewer for CF-I mediated regulation of 3'UTRs, and thank the reviewer for this insightful comment.

We agree that successful rescue experiments will further strengthen our conclusion that downregulation of CF-I contributes to 3'UTR shortening during muscle differentiation. That said, reversing 3'UTR shortening by re-expression of CFI-25 and CFI-68 in myotubes could be very tricky. First, due to low transfection/infection efficiency of myotubes, we will need to do overexpression in myoblasts before differentiation. However, overexpression of CFI-25 and CFI-68 in myoblasts may disrupt normal polyadenylation processes during proliferation and cause unforeseen effect. Second, we will need to co-express CFI-25 and CFI-68 as they need to assemble into a tetramer for action. This might be also challenging for achieving balanced stoichiometric levels, as unbalanced expression of exogenous CFI-25 and CFI-68 might act as molecular competitors to interfere with endogenous CF-I complexes.

The reviewer also suggesting direct binding assays to show CF-I binding on its APA-regulated substrate RNAs. In response, we analyzed public CFI-68 tRIP-seq data (Kawachi *et al*, 2021) (a method analogous to CLIP-seq) in myoblasts to evaluate CFI-68 binding across different APA substrate RNAs. We observed that mRNAs undergoing 3'UTR shortening showed stronger CFI-68 binding compared to those with lengthened or unchanged 3'UTR. Furthermore, when we categorized the shortened mRNAs into two groups depending on whether CFI-68 KD led to their 3'UTR shortening or not, we found that the responsive mRNAs exhibited higher CFI-68 binding than those not response to CFI-68 downregulation. These findings strengthen the role of CFI in regulating 3'UTR shortening during myogenesis. We have now incorporated these new results in the revised manuscript (**revised Fig. 3D; Page 9, Lines 202-205**).

5) It would be useful to provide some discussion on why CFI expression is downregulated.

As suggested, we have included discussion regarding the potential mechanisms underlying CFI downregulation in the revised manuscript (**Pages 17-18, Lines 421-429**).

6) A greater binding of Ago2 to aUTRs of 3'UTR shortened genes is interesting. On the one hand it may indicate shortening of 3'UTR can evade miRNA targeting. On the other, it suggests that some of the shortening cases may be attributable to long isoform degradation. The authors need to discuss the possibility of latter and whether it may also contribute the APA isoform difference between different models, i.e., distinct miRNA targeting activities in different systems may impact the 3'UTR length profile differently.

We agree with the reviewer that 3'UTR shortening events observed in whole cells might partly stem from degradation of the long isoform. As suggested, we have discussed this possibility in the revised manuscript (**Pages 5-6, Lines 117-132**). We would like to note that to exclude this possibility, we focused on the events that were observed both in whole cells and in the nuclear fraction.

7) Further on the discussion of the difference between C2C12 model and primary muscle stem cells, the authors may want to comment on the differentiation stages, e.g., early vs. late. Pertinent to myogenesis, there is a cell fusion step during differentiation. Would it be possible that the 3'UTR length control is different before vs. after this step?

We thank the reviewer for raising this important point. In response, we have expanded our discussion on this point in the revised manuscript (**Page 17, Lines 411–416**).

As the reviewer rightly noted, cell fusion from mononucleated myoblasts to multinucleated myotubes is a key step in myogenesis. Indeed, it is plausible that APA dynamics differ before and after fusion. In our experimental setup, however, samples were harvested at the terminal differentiation stage when cell fusion is largely complete. We did not find suitable publicly transcriptomic data to explore this possibility through a search of the literature. Future studies employing time-resolved profiling of APA dynamics across the entire differentiation process could help elucidate whether distinct patterns of proximal versus distal polyA site usage are linked to pre-fusion and post-fusion stages.

Referee #2:

In this manuscript, Zhu et al characterized mRNA alternative polyadenylation (APA) during myogenic differentiation, and elucidated the functional significance of key genes that displayed significant APA changes. Although there were previous studies that looked into this topic using cell line systems, this study used stem cells isolated from mice and was therefore more physiologically relevant and found new and interesting targets. Additionally, they not only examined the molecular mechanisms for the observed APA changes, but also meticulously investigated the downstream functional impact using a variety of approaches, from antisense oligoes to mutant animal models. It is quite rare and refreshing to see such in-depth and comprehensive studies. I think this study is fully ready to be published in EMBO J as is.

We are grateful for the reviewer's appreciation of our work.

Referee #3:

In the manuscript by Zhu, Wang, Tong, Jia et al. entitled "3'UTR shortening alleviates miRNA repression for muscle stem cell differentiation" the authors explore the interplay between 3'UTR length and targeting by muscle-enriched miRNAs (myomiRs) during muscle cell differentiation. They find results different than previous studies with cell lines and validate their results in vivo. This work provides new insights into the intricacies of muscle cell differentiation and repair.

The study includes a number of complementary approaches to validate the results obtained, including the generation and analysis of a mouse model to validate the findings in vivo with a focus on one key gene identified in their analysis, Matr3. These orthogonal approaches provide mechanistic insight.

We thank this reviewer for his/her appreciation for our work and for the helpful comments.

1) As a rather minor point, the authors should carefully consider their language throughout as no genes are shortened or lengthened- the transcripts, or more precisely the 3'UTR of the transcripts, are shortened or lengthened. The use of the term shortened gene is short-hand to indicate that the gene encodes a transcript with an altered 3'UTR. The authors should just carefully consider the precision of the language used throughout. For example, in Figure 4I, the authors should really change the label as it currently reads "Shortened genes with increased Ago2 binding".

The genes are not shortened and the Ago2 is actually not binding to genes.

We thank the reviewer for this important point. As suggested, we have thoroughly revised the manuscript to ensure precise descriptions regarding 3'UTR length changes. Imprecise labels in the figures and figure legends, as well as relevant descriptions in the main text, have been corrected in the **revised manuscript and figures**.

2) The statistical tests employed to analyze the data are not described in sufficient details in either the Materials and Methods or the Figure Legends. The authors need to clearly indicate what type of statistical tests are performed for each analysis. Although p-values are provided throughout, the type of tests performed (appears primarily to be t-tests) are not indicated.

We agree that the statistical methods were not described in sufficient detail. In response, we have updated the manuscript and specified the type of statistical test used for each analysis in the corresponding **figure legends**.

*3) In addition, results that are statistically significant are not indicated by asterisks as is typical. This makes the data more difficult for the reader to interpret. The authors should employ the conventional approach of indicating statistical significance and degree of statistical significance with *, **, ***.*

Thank you for this suggestion. Regarding the indication of statistical significance, to ensure a more quantitative representation of statistical strength, we would like to keep the exact p-values for all analyses, allowing readers to better evaluate the robustness of the findings.

4) For the analysis shown in Figure 1C, the KEGG analysis of genes where the transcripts are shortened, the authors identify key categories critical for muscle differentiation. The text does not make clear whether the authors also performed a similar analysis with the set of genes that have lengthened 3'UTRs or those that do not change. As these are primary muscle cells, the terms identified are certainly likely to relate to muscle. The most robust analysis would focus on those terms enriched for the transcripts that show a shortened 3'UTR compared to other datasets- not just the shortened 3'UTR transcripts as an independent dataset. Perhaps the authors performed this analysis, but the point is not clear in the text or the figure.

As suggested, we have performed KEGG enrichment analysis for other two groups, lengthened and unchanged 3'UTRs. Although these two groups of genes also

showed enrichment in muscle-related pathways, the extent was much lesser than shortened 3'UTR genes, as indicated by fewer terms, fewer gene counts, and less significant adjusted p-values. We have included these results in the revised manuscript (**revised Fig. EV1A; Page 5, Lines 108-112**).

5) *Figure 2A, which indicates the reads for the Matr3 gene does not indicate the number of bases on the X-axis. This information should be included in the figure as shown markers on the gel image shown in Figure 2B. In Figure 2G and other such figures, labeling the Y-axis as 'Relative Transcript Abundance' would make the figure easier to interpret for the reader along with the main text of the manuscript without having to consult the figure legend, which does clearly indicate the analysis of transcript levels.*

Thanks. As suggested, we have added genomic scale markers and updated Y-axis labels in all-related figures.

6) *The data presented in Figure 2H-K does not seem to include the transcripts where the ASO treatment did NOT impair MT formation (beyond the proof of principle in Figure 2H that the ASO works to shift the ratio of short to long (S/L) 3'UTR in Figure 2H. Instead, these controls are shown in Figure S2, but they should be incorporated into the primary data shown in the main text figures.*

Thank you for this valuable suggestion. As suggested, we have now incorporated the relevant data into the main figure (**revised Fig. 2I-2L**).

7) *For the broad readership of EMBO J., the authors should be sure to define all abbreviations including in the figures. Figure 1A includes the abbreviations SC and MT without definition in the figure (where there really is space) or, at a minimum, in the Figure Legend.*

Thank you for this helpful suggestion. We have now defined the abbreviations in the **revised Fig. 1A, 1E and 4B**.

8) *The formatting of one of the sets of references on p. 8 has an error because the title of the one of the papers is included in the list of references.*

We thank the reviewer for pointing this out. We have now corrected it in the revised manuscript.

Referee #4:

In this study, the investigators provided some interesting findings to show the 3' UTR shortening is associated with gene function and muscle differentiation.

We thank the reviewer for pointing out that our findings are interesting.

1) When performing polyA RNA-seq, why they separated nucleus and cytoplasm portion? The difference in 3'UTR shortening is quite different in N. and C. compartments. Why? Why were these results so different from that of the QuantSeq analysis? (Fig. 1).

We thank the reviewer for these insightful questions regarding our experimental approach and results. We performed RNA-seq in the nuclear and cytoplasmic fractions for two reasons. First, we sought to analyze nuclear export regulation for mRNAs exhibiting different 3'UTR length changes. Second, we aimed to distinguish genuine APA regulation that occurs in the nucleus from potential cytoplasmic effects, such as preferential degradation of long 3'UTR isoforms in the cytoplasm.

Our polyA RNA-seq data revealed pronounced 3'UTR shortening in both the nuclear and cytoplasmic fractions, supporting the conclusion that increasingly used proximal polyA site during satellite cell differentiation. We would like to note that in the cytoplasmic fraction, the ratio of 3'UTR-shortened genes to lengthened genes is approximately 5:1 (1531/303), whereas the ratio was 2:1 (799/412) in the nuclear fraction. This difference may be partly due to preferential degradation of long isoforms in the cytoplasm, such as miRNA-mediated silencing, and/or preferential nuclear export of short isoform (Bartel, 2018; Chen *et al*, 2019; Tang *et al*, 2022; Neve *et al*, 2016).

The quantitative differences observed between PolyA RNA-seq and QuantSeq likely stem from their methodological distinctions. QuantSeq, which primarily captures reads near the 3' end of transcripts, offers much higher sensitivity in detecting polyA site usage. Meanwhile, it might detect rarely used polyA site, including which might cause false-positive changes. In contrast, PolyA-seq profile full-length transcripts, which can lead to less uniform coverage of 3' termini and may consequently be less effective at resolving subtle APA variations. Furthermore, the distinct computational pipelines (QuantSeq-MAAPER vs. polyA RNA-seq-APAlyzer) applied to each

dataset may also contribute to the quantitative disparities, as they rely on different statistical models. We would like to note that both methods led to same conclusion that 3'UTRs tend to be shortened during myogenesis.

2) The authors cited Quintero-Rivera (Human Molecular Genetics 24: 2375-2389), and stated that "its short 3'UTR isoform is specifically detected in skeletal and cardiac muscle tissues", and "truncated Matr3 3'UTR exhibited abnormal motor functions". These statements are not true. The prior study described that "Matr3(Gt-ex13) heterozygotes exhibit incompletely penetrant BAV, CoA and PDA phenotypes similar to those in the human proband, as well as ventricular septal defect (VSD) and double-outlet right ventricle (DORV)". Those are not muscle expressing tissues and there is no defect in motor function.

Thank you for this critical comment. Upon re-examination of the study by Quintero-Rivera et al., we confirmed that the different 3'UTR isoform of *Matr3* was indeed detected in heart, skeletal muscle and other tissues.

The proband of this study presented with a spectrum of congenital heart defects, including BAV, CoA, PDA, VSD, and DORV. Furthermore, the Supplementary Information of their study reported impaired fine motor skills (**quoted** from (Quintero-Rivera *et al*, 2015): "At ten years of age, he could use 50 words and 50 signs to communicate, and continued to have difficulty with fine motor skills"). We acknowledge that our original description of "abnormal motor functions" might overemphasized this particular impairment, potentially leading to misinterpretation among readers. Accordingly, we have modified the phrasing to "severe cardiac defects and impaired fine motor skills" to better reflect the reported findings in the revised manuscript (Page 7, Line 145).

3) They verified the 3'UTR shorting of *Matr3* gene in C2C12 myoblast differentiation. They should do that for their other tested genes (*Cpd*, *Foxp1*, *Il6st*, *Neo1*, *RalA*, and *Sec63*) as well.

Thank you for this helpful suggestion. As suggested, we have now examined the 3'UTR changes of other candidate genes during myoblast differentiation. Consistent with our sequencing data, these gene also showed an increased S/L isoform ratio in myotubes compared to myoblast. These results are now presented in **revised Fig. EV2D**.

4) Fig. 3I, they used RT-PCR to document *Luc-S/Luc-L* ratio. Can they also provide quantitative measurement of the luciferase activities?

Thank you for this suggestion. We would like to clarify that the goal of this experiment was to quantitatively assess the role of UGUA/CFI in APA regulation of *Matr3*. Since the luciferase activity assay cannot specifically distinguish between the two isoforms generated from proximal versus distal polyA sites, we used isoform-specific RT-qPCR to accurately quantify the relative usage of each polyA site. To avoid any potential confusion, we have relabeled the relevant figures (**revised Fig 3I**).

5) They found that "genes with lengthened or unchanged 3'UTRs, but not that of shorted, showed significant decreases in polysome association in MTs versus MBs (Figure 4)". It will be nice if they can show protein levels of representative targets.

Thank you for raising this point. We would like to note that, in addition to translation regulation, changes in other aspects of mRNA and proteins levels during muscle differentiation could influence final protein abundance as well. Given these multifactorial regulatory layers, it would be challenging to directly correlate altered polysome association with corresponding changes in protein expression, and this complexity also limited our selection of suitable representative targets.

6) Wasn't the AGO2 CLIP data from Zhang 2014 showed Ago2 binding to 3'UTRs of mitochondrial mRNAs?

As the reviewer said, the study by Zhang et al. (2014) focused on Ago2 binding on

mitochondrial mRNAs. Given that CLIP-seq data detect global Ago2 binding profiles, we analyzed its binding on mRNAs beyond mitochondrial mRNAs in myoblasts and myotubes.

7) *The muscle phenotype data from the $Matr3^{mpA/mpA}$ (without injury) is not consistent with that of in vitro study, in which they showed that $Matr3$ -pAMO affected SC differentiation. How do you reconcile this? Is $matr3$ (or different isoforms of the transcript) expressed in SCs, undifferentiated MBs and differentiated MTs with specificity?*

We thank the reviewer for raising this point. We reason the phenotypic differences between uninjured $Matr3^{mpA/mpA}$ mice and $Matr3$ -pAMO-treated SC cells can be attributed to differences in differentiation/regenerative demand.

Under homeostatic conditions (without injury), where regeneration demand is low, $Matr3^{mpA/mpA}$ mice do not exhibit overt muscle abnormalities. This observation suggests that the reduced differentiation/regeneration capacity and/or compensatory mechanisms inherent to the muscle niche are sufficient to sustain normal postnatal muscle development. This aligns with findings for other key muscle regulatory genes: for example, *Pax7* and *MyoD* knockout mouse models display mild developmental defects, despite their critical roles in satellite cell self-renewal and differentiation (Oustanina *et al*, 2004; Rudnicki *et al*, 1992).

In contrast, in the context of CTX-induced muscle injury that requires robust satellite cell regenerative capacity, $Matr3^{mpA/mpA}$ mice exhibited clear muscle regeneration defects (**Fig. 6D, E**). This impairment was further exacerbated following repeated injury challenges (**Fig. 6F, G**), underscoring the requirement for *Matr3* 3'UTR shortening in sustaining robust regenerative output. Collectively, these results reveal that *Matr3* 3'UTR shortening functions in satellite cells responses to regenerative demand, with its role becoming particularly essential under high-demand conditions.

Similar to high regenerative demand during muscle regeneration after injury, *in vitro* differentiation of satellite cells isolated from muscle also requires robust myoblast differentiation. Indeed, we observed impaired myotube formation in satellites isolated from $Matr3^{mpA/mpA}$ mice (without injury) (**Fig. 6H-J**). Notably, this impairment is consistent with the phenotype observed in normal satellite cells treated

with *Matr3*-pAMO.

Regarding the expression of *Matr3* isoforms, we found that the long isoform of *Matr3* is highly expressed in proliferating satellite cells and myoblasts, whereas the short isoform becomes predominant in terminally differentiated myotubes (**Figs. 2B and EV2A**).

8) *They have used antisense morpholino oligonucleotide to block proximal pA site of the Matr3 transcripts. As we all know AMOs could be non-specific, or off-target, they should utilize an independent assay to verify their findings.*

The reviewer raised a concern regarding the potential off-target effects of AMO targeting the pPAS of *Matr3*, and suggest to verify the findings on the role of *Matr3* 3'UTR shortening in myogenic differentiation. We would like to emphasize that in the *Matr3*^{mpA/mpA} mutant mice we generated, the pPAS was mutated, thereby abolishing the 3'UTR shortening. When satellite cells isolated from *Matr3*^{mpA/mpA} mice were induced to differentiate *in vitro*, they recapitulated the differentiation defect observed upon *Matr3*-pAMO treatment (as shown in **Fig. 6H-6J**). This genetic model-based evidence, which is independent of AMO-based perturbation, further verifies the functional importance of *Matr3* 3'UTR shortening in myogenic differentiation.

9) *The data presented in Fig 7 is very confusion. Muscle defect in DMD patients was caused by mutations of the dystrophin gene. How do those dystrophin mutations then lead to dysregulation of 3'UTR shorting?*

We thank the reviewer for this insightful comment. We totally agree with the reviewer that dystrophin deficiency itself is unlikely to disrupt APA directly. However, it is possible that the pathological muscle microenvironment in DMD disease may lead to dysregulated 3'UTR shortening. Such dysregulation may subsequently influence disease progression by modulating the expression of genes critical for muscle function. In response to this comment, we have discussion this possibility into the revised manuscript to better clarify our proposal in the revised manuscript (**Page 15-16, Lines 375-379**).

10) *The hypothetic models presented in Figs 7 and 8 are quite confusion, and overinterpreted their findings. The roles of myomiRs in muscle differentiation was*

never tested directly in the context of *Matr3* 3'UTR. They only presented that in the luciferase reporter assays performed in non-muscle 293T cells.

The reviewer raised a point regarding Figs 7 and 8.

Our intention in Fig 7 was to propose a plausible link between 3'UTR shortening and its potential relevance in muscle-related diseases. We agree that some descriptions in the original manuscript may have overinterpreted the findings. In response to your concern, we have carefully revised the relevant descriptions in the Fig. 7 section (**Pages 15-16**) to more accurately reflect the observations and proposals of this study.

For Fig 8, we would like to note that in addition to the luciferase reporter data from HEK293 cells, we provided two lines of experimental indicating miRNA-mediated regulation of *Matr3* and its functional in muscle cells:

- (1) Luciferase reporter assays in myoblasts and myotubes confirmed miR-1/206 targeting of the *Matr3* 3'UTR (**Fig. 5G**);
- (2) Using a target-site-specific AMO to block the miR-1/206 targeting of the *Matr3* 3'UTR (*Matr3*-tss AMO), we showed that miRNA-mediated regulation is important for regulating *Matr3* protein levels and supporting SC differentiation (**Figs. 5H–5K**).

11) *Incorrect formatting or incomplete in reference citations:(Tian & Manley, 2017; Lackford et al, 2014; Li et al, 2015; The Polyadenylation Factor CstF-64 Regulates Alternative Processing of IgM Heavy Chain Pre-mRNA during B Cell Differentiation, 1996; Martin et al, 2012)*

Thanks. We have corrected this in the revised manuscript.

Reference:

- Bartel DP (2018) Metazoan MicroRNAs. *Cell* 173: 20–51
- Chen S, Wang R, Zheng D, Zhang H, Chang X, Wang K, Li W, Fan J, Tian B & Cheng H (2019) The mRNA Export Receptor NXF1 Coordinates Transcriptional Dynamics, Alternative Polyadenylation, and mRNA Export. *Molecular Cell* 74: 118-131.e7
- Kawachi T, Masuda A, Yamashita Y, Takeda J, Ohkawara B, Ito M & Ohno K (2021) Regulated splicing of large exons is linked to phase-separation of vertebrate transcription factors. *EMBO J* 40: e107485
- Neve J, Burger K, Li W, Hoque M, Patel R, Tian B, Gullerova M & Furger A (2016) Subcellular RNA profiling links splicing and nuclear DICER1 to alternative cleavage and polyadenylation. *Genome Res* 26: 24–35

- Oustanina S, Hause G & Braun T (2004) Pax7 directs postnatal renewal and propagation of myogenic satellite cells but not their specification. *EMBO J* 23: 3430–3439
- Quintero-Rivera F, Xi QJ, Keppler-Noreuil KM, Lee JH, Higgins AW, Anchan RM, Roberts AE, Seong IS, Fan X, Lage K, *et al* (2015) MATR3 disruption in human and mouse associated with bicuspid aortic valve, aortic coarctation and patent ductus arteriosus. *Human Molecular Genetics* 24: 2375–2389
- Rudnicki MA, Braun T, Hinuma S & Jaenisch R (1992) Inactivation of MyoD in mice leads to up-regulation of the myogenic HLH gene myf-5 and results in apparently normal muscle development. *Cell* 71: 383–390
- Tang P, Yang Y, Li G, Huang L, Wen M, Ruan W, Guo X, Zhang C, Zuo X, Luo D, *et al* (2022) Alternative polyadenylation by sequential activation of distal and proximal PolyA sites. *Nat Struct Mol Biol* 29: 21–31

Prof. Hong Cheng
Shanghai Institute of Biochemistry and Cell Biology
320 Yue Yang Road
Shanghai 200031
China

14th Nov 2025

Re: EMBOJ-2025-121889R
3'UTR shortening alleviates miRNA repression for muscle stem cell differentiation

Dear Hong,

Thank you again for submitting your revised manuscript, and my sincere apologies for the additional delays with its re-evaluation. Of the original reviewers, only referee 3 was available to look through your responses and revisions, but given their positive assessment, I decided to eventually proceed further towards pre-acceptance of the study. Prior to publication, there remain only a number of editorial issues, which I would ask you to incorporate during a final round of minor revision:

- I am wondering whether the title could be slightly expanded to make it a bit more explicit and accessible also for a broader non-expert readership - e.g.

3'UTR shortening alleviates miRNA repression of mRNAs critical for muscle stem cell differentiation

Happy to discuss!

- On the abstract page of the manuscript, please include 4-5 general keyword terms to enhance searchability.

- Please carefully go through the reference list and make sure that each reference is complete with citation year, volume, and page/eLocator numbers - this information is currently missing for several of them (and one of them, Kang 2023b, also appears strangely formatted).

- As we are switching from a free-text author contribution statement towards a more formal statement based on Contributor Role Taxonomy (CRediT) terms, please remove the present Author Contribution section and instead specify each author's contribution(s) directly in the Author Information page of our submission system during upload of the final manuscript. See <https://casrai.org/credit/> for more information.

- In the Data Availability section, please remove the referee access information, and ensure that the data are becoming publicly available at this point. Furthermore, please note that only newly derived/deposited datasets should be listed here. For previously reported/deposited datasets, please mention them were appropriate (e.g. in the Results and/or Methods section), and consider including formal Data Citations (explained at: <https://www.embopress.org/page/journal/14602075/authorguide#referencesformat>)

- Please double-check and correct all reference for data panels in the text (e.g. Appendix Figure S2 panels are currently only called out as "Appendix S2").

- For the EV Datasets, please make sure that their respective title/legend/description is always included in a separate "Legend" tab of the XLSX spreadsheet, rather than in a separate README file. These files should be uploaded in their primary format (e.g. DOC, XLS), not as archives.

- Finally, during routine pre-acceptance checks, our data editors have raised the following queries regarding figures, data, and legends; I would appreciate if you briefly answered to them in the cover letter of your final submission, and made the requested text modifications with changes/additions highlighted via the "Track changes" option, to facilitate our final checking"

1. Please note that the exact p values need to be provided in the legends of figures 1D, 2F, G, K; 3C, D, G.
2. Please indicate the statistical test used for data analysis in the legends of figures 1C, 7B, EV1 A, B; EV4 A, B.
3. Please note that the box plots need to be defined in terms of minima, maxima, centre, bounds of box and whiskers, and percentile in the legends of figures 3C, D; 4A
4. Please note that the box plots need to be defined in terms of minima, maxima, bounds of box and whiskers, and percentile in the legends of figures 1D
5. Please note that information related to n is missing in the legends of figures 1D, 3C, D; 4A, C
6. Please note that the error bars are not defined in the legend of figure 4C

I am returning the manuscript to you for a final round of minor revision, solely to allow you to make these modifications and upload the revised files. Once we will have received them, we should be ready to swiftly proceed with formal acceptance and production of the manuscript.

With kind regards,

Hartmut

9) To facilitate reproducibility and cross-laboratory adoption of methodologies, please structure the Materials & Methods section as outlined in our guide to authors, including a completed Reagents and Tools Table that can be downloaded from our author guidelines as well (<https://www.embopress.org/page/journal/14602075/authorguide#structuredmethods>).

10) Digital image enhancement is acceptable practice, as long as it accurately represents the original data and conforms to community standards. If a figure has been subjected to significant electronic manipulation, this must be clearly noted in the figure legend and/or the 'Materials and Methods' section. The editors reserve the right to request original versions of figures and the original images that were used to assemble the figure. Finally, we generally encourage uploading of numerical as well as gel/blot image source data; for details see: embopress.org/page/journal/14602075/authorguide#sourcedata

In the interest of ensuring the conceptual advance provided by the work, we recommend submitting a revision within 3 months (12th Feb 2026). Please discuss the revision progress ahead of this time with the editor if you require more time to complete the revisions. Use the link below to submit your revision:

Link Not Available

Referee #3:

In the manuscript by Zhu, Wang, Tong, Jia et al. entitled "3'UTR shortening alleviates miRNA repression for muscle stem cell differentiation" the authors explore the interplay between 3'UTR length and targeting by muscle-enriched miRNAs (myomiRs) during muscle cell differentiation. They find results different than previous studies with cell lines and validate their results in vivo. This work provides new insights into the intricacies of muscle cell differentiation and repair.

The study includes a number of complementary approaches to validate the results obtained, including the generation and analysis of a mouse model to validate the findings in vivo with a focus on one key gene identified in their analysis, *Matr3*. These orthogonal approaches provide mechanistic insight.

In this revised version of the manuscript, the authors have thoroughly considered the reviewer comments, added new data, and edited the manuscript to address the points raised by the reviewers.